# Batch normalization followed by merging is powerful for phenotype prediction integrating multiple heterogeneous studies

**Yilin Gao**[ID], **Fengzhu Sun**[ID]*

Department of Quantitative and Computational Biology, University of Southern California, Los Angeles, California, United States of America

* fsun@usc.edu

## Abstract

Heterogeneity in different genomic studies compromises the performance of machine learning models in cross-study phenotype predictions. Overcoming heterogeneity when incorporating different studies in terms of phenotype prediction is a challenging and critical step for developing machine learning algorithms with reproducible prediction performance on independent datasets. We investigated the best approaches to integrate different studies of the same type of omics data under a variety of different heterogeneities. We developed a comprehensive workflow to simulate a variety of different types of heterogeneity and evaluate the performances of different integration methods together with batch normalization by using ComBat. We also demonstrated the results through realistic applications on six colorectal cancer (CRC) metagenomic studies and six tuberculosis (TB) gene expression studies, respectively. We showed that heterogeneity in different genomic studies can markedly negatively impact the machine learning classifier's reproducibility. ComBat normalization improved the prediction performance of machine learning classifier when heterogeneous populations are present, and could successfully remove batch effects within the same population. We also showed that the machine learning classifier's prediction accuracy can be markedly decreased as the underlying disease model became more different in training and test populations. Comparing different merging and integration methods, we found that merging and integration methods can outperform each other in different scenarios. In the realistic applications, we observed that the prediction accuracy improved when applying ComBat normalization with merging or integration methods in both CRC and TB studies. We illustrated that batch normalization is essential for mitigating both population differences of different studies and batch effects. We also showed that both merging strategy and integration methods can achieve good performances when combined with batch normalization. In addition, we explored the potential of boosting phenotype prediction performance by rank aggregation methods and showed that rank aggregation methods had similar performance as other ensemble learning approaches.

**Data Availability Statement:** All the CRC datasets used in this study are publicly available in the European Nucleotide Archive (ENA) database (https://www.ebi.ac.uk/ena). Accession number for Yu is PRJEB10878, for Hannigan is PRJNA389927,

for Feng is ERP008729, for Vogtmann is PRJEB12449, for Zeller is ERP005534, and for Thomas is SRP136711. All the TB annotated and cleaned datasets used in this study are obtained from the study by Zhang et al. and available on their github repository: https://github.com/zhangyuqing/bea_ensemble All the codes used in analysis can be found at https://github.com/lynngao/Heterogeneous-Studies.

**Funding:** This research was supported by US National Institutes of Health [1R01GM131407] and the National Science Foundation [EF-2125142] to FZS, which supported the work of FZS and YLG. The funders had no role in study design, data collection and analysis, decision to publish, or preparation of the manuscript.

**Competing interests:** The authors have declared that no competing interests exist.

## Author summary

Overcoming heterogeneity when incorporating different studies in terms of phenotype prediction is a challenging and critical step for developing machine learning algorithms with reproducible prediction performance on independent datasets. We developed a comprehensive workflow to simulate a variety of different types of heterogeneity and evaluate the performances of different integration methods together with batch normalization by using ComBat. We also demonstrated the results through realistic applications on six colorectal cancer (CRC) metagenomic studies and six tuberculosis (TB) gene expression studies, respectively. From both the simulation studies and realistic applications, we showed that batch normalization is essential for improving phenotype prediction performance by machine learning classifiers when incorporating multiple heterogeneous datasets. Combined with batch normalization, merging strategy and ensemble weighted learning methods both can boost machine learning classifier's performance in phenotype predictions. In addition, we explored that rank aggregation methods should be considered as alternative ways to boost prediction performances, given that these methods showed similar robustness as ensemble weighted learning methods.

## Introduction

Genotype to phenotype mapping is an essential problem in the current genomic era. With the development of advanced biotechnologies, many types of genomic data such as single nucleotide polymorphisms, gene expression profiles, proteomics, metagenomics, etc. have been generated in many different studies. These omics data provide essential resources to understand the relationships between omics data and phenotypes. Despite these fundamental developments, due to the heterogeneity of data, it is challenging to integrate the omics data to understand genotype to phenotype mapping. For a single type of data such as gene expression or metagenomic data, many sources of heterogeneity can occur. For example, the samples can come from different ethnic groups with varying underlying distributions of the features. Even if the samples come from the same population, the genomic data can be generated from different laboratories and/or derived from different experimental technologies resulting in different distributions of the data. Another types of heterogeneity can be caused by the different causal mechanisms of the same phenotype in the populations under study [1]. The objective of this study is to investigate the best approaches to integrate different studies of the same type of data under a variety of different heterogeneities. In this work, we concentrate on gene expression profiles or microbial abundance in metagenomic studies.

Many machine learning algorithms including linear regression, logistic regression, penalized regression, support vector machines (SVM), random forests (RF), neural networks (NN) and deep neural networks (DNN) have been used to predict phenotypes from omics data [2–5]. Most previous studies validated the prediction methods using within dataset cross validation usually with relatively high prediction accuracy. However, the prediction accuracy is markedly decreased when the learned algorithms are used in independent datasets [6, 7]. Many sources of study heterogeneity, for example, different experimental platforms or procedures and differences in patient cohorts [1], all contribute to compromise the prediction performance of machine learning models in cross-study settings. Thus, overcoming heterogeneity in cross-study phenotype prediction is a critical step for developing machine learning algorithms with reproducible prediction performance on independent datasets.

Many studies have been carried out to mitigate the heterogeneity in cross-study phenotype predictions. Zhang et al. [8] focused on the batch effects of data when developing genomic classifiers. Patil et al. [9] simulated genomic samples and perturbed the coefficients of linear relations between outcomes and predictors to evaluate model reproducibility with different degrees of heterogeneity. In this study, we address three types of heterogeneity: different background distributions of genomic features in populations, batch effects across different studies from the same population, and different disease models in various studies. We aim to evaluate how different statistical methods can mitigate these three types of heterogeneity.

Merging all datasets into one and treating all samples as if they are from the same study is a generally used method for cross-study predictions. With the increase of sample size and diversity in the study population, merging method has been shown to lead to better prediction performance than using only individual studies [2, 5, 10]. Another approach is to integrate the trained predictors from different machine learning models derived from various training datasets. Ensemble weighted learning is a commonly used integration method to deal with the impact of heterogeneity on cross-study prediction performance. Ensemble learning methods that integrate predictions from multiple machine learning models showed the ability to boost the prediction performance than using only the component methods that the ensemble learning contains [9, 11]. Besides ensemble weighted learning methods, aggregating the ranks from sample predicted probability instead of the probability itself offers a promising alternative for integration. In some situations, the predicted probabilities for the samples in the test data may not be correct, but the relative order could provide some useful information. In such situations, aggregating the ranks instead of the predicted probabilities might be more reasonable. To the best of our knowledge, no studies investigated rank aggregation methods based on omics data phenotype prediction.

ComBat [12] normalization is a commonly used method for removing batch effects between different datasets. In our previous study [13], we showed that when dealing with heterogeneity in cross-study predictions, applying ComBat only before training machine learning models did not improve the prediction performance. Zhang et al. [8] showed that ensemble weighted learning methods outperform batch correction by ComBat at high level of batch differences. Nevertheless, in this study, we aim to explore the potential of combining the normalization of ComBat together with merging and integration methods (ensemble weighted learning and rank aggregation) in the presence of three different types of heterogeneity mentioned above. We provide both simulations and real data applications on metagenomic and gene expression data to show the comparisons of performance from different statistical methods when dealing with cross-study heterogeneity.

Our study offers innovations through the development of a comprehensive workflow that effectively addresses heterogeneity in various types of omics data, an issue that has persistently compromised the performance of machine learning models in cross-study phenotype predictions. By investigating the optimal approaches to integrate different studies and conducting realistic applications on colorectal cancer and tuberculosis studies, we have created a robust framework that elevates the reproducibility of machine learning classifiers. Furthermore, the innovative utilization of ComBat normalization with integration methods markedly enhances prediction performance through removing heterogeneity effects. Our study also ventures into mostly uncharted territory by exploring the potential of boosting phenotype prediction performance through rank aggregation methods in handling heterogeneous omic data, a significant innovation in its own right.

## Methods

### Outline of workflow for integrating multiple simulated heterogeneous metagenomic datasets

To investigate the prediction performance of different merging and integration methods when applied to multiple heterogeneous datasets, we developed a comprehensive workflow with three main steps to conduct the experiments.

The first step in our workflow involves simulating diverse metagenomic datasets under three distinct scenarios, as illustrated in Fig 1A. In the first scenario, we examine how divergent training and test populations with varied genomic feature distribution can impact the predictive performance of machine learning models.

Population differences, such as variances in ethnicity, diet, and other factors, can lead to variations in the genomic features within a population. These features encompass single nucleotide polymorphisms (SNPs), expression levels, and microbial abundance in the microbiome. Consequently, when training machine learning classifiers on one population and applying them to a different population, it is crucial to account for the heterogeneity in the background distributions. Neglecting this heterogeneity can adversely affect the prediction performance.

To assess how heterogeneity arising from population differences can influence the performance of machine learning classifiers, and to identify the optimal approaches for integrating prediction methods from various heterogeneous studies, we simulated three distinct populations. Each population was characterized by different background genomic distributions, and we manipulated these differences. For a detailed description of the implementation process for simulating the two training datasets and one test dataset, please refer to Scenario 1: Divergent training and test populations with varied genomic feature distribution.

The second scenario of heterogeneity pertains to batch effects. In simple terms, batch effects represent non-biological disparities that emerge across separate batches of data. These discrepancies are usually a byproduct of technical variances in experimental conditions or different labs. Some contend that batch effects can compromise the replicability of genomic studies [14].

It is commonplace to correct these batch effects in the pre-processing phase when dealing with genomic data. Recently, several methods for batch effect correction have been proposed, including ComBat [12], edgeR [15], and DESeq2 [16]. Additionally, some studies have suggested the use of ensemble learning techniques to potentially minimize batch effects [8, 9].

In our study, we sought to evaluate these methods' efficacy in reducing batch effects on the predictive performance of binary classifiers. This was achieved through simulation studies. In this particular setup, the training and test datasets originate from the same genomic feature distribution. However, upon simulating the training and test datasets, we then introduced batch effects into the training datasets, creating two distinct batches. These two batches were then used as the training datasets for the subsequent experiments. We provide a detailed description of the method in Scenario 2: Different batch effects on training data with consistent underlying population genomic feature distribution.

In the aforementioned two scenarios, we examined how both the variability in the distribution of genomic features among populations and batch effects influence a classifier's performance. Furthermore, we operated under the assumption that the disease models are consistent across either the same or different populations. However, a range of studies have suggested that the microbes linked with certain diseases can depend on the population in question. For instance, research indicates that colorectal cancer development can vary among populations with differing rates of diabetes [17], smoking [18], and obesity [19].

Consequently, we have further assessed the effectiveness of merging and various integration methods when the underlying disease models differ. More specifically, the degree of overlap in

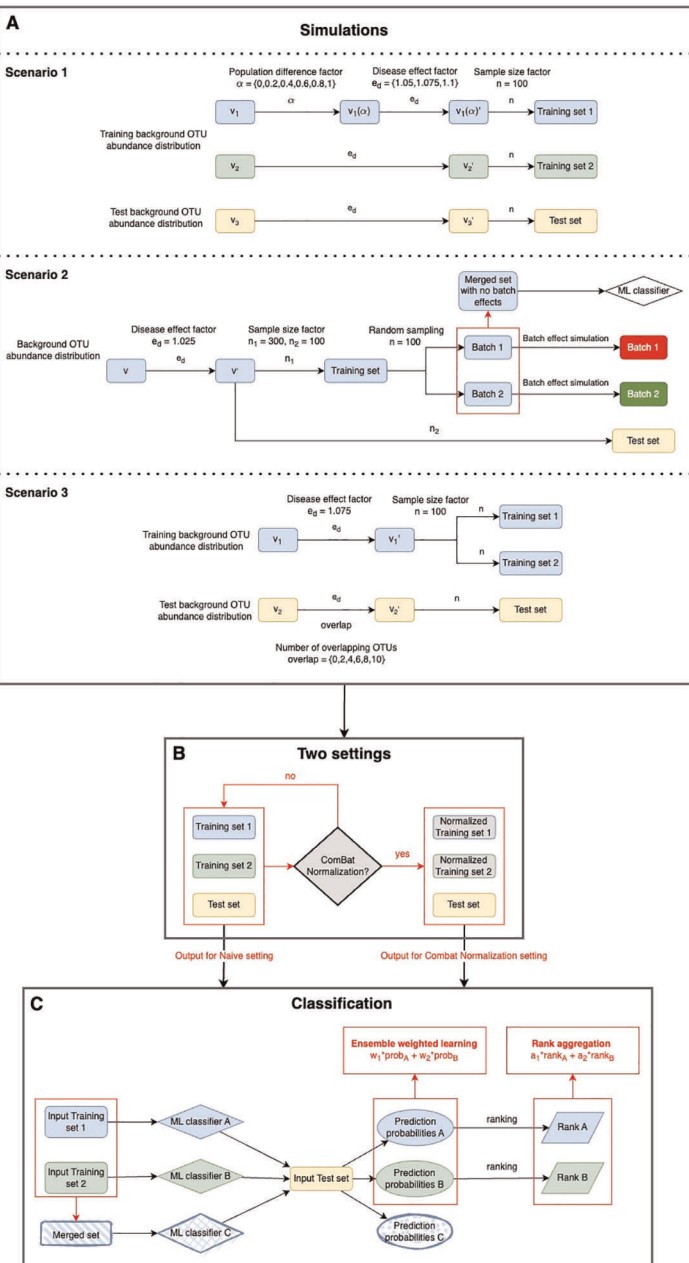

**Fig 1. Workflow for integrating multiple simulated heterogeneous metagenomic datasets.** A: Simulation stage of three different heterogeneity scenarios. Scenario 1: Divergent training and test populations with varied genomic feature distribution. Scenario 2: Different batch effects on training data with consistent underlying population genomic feature distribution. Scenario 3: Varying degrees of overlap in disease-associated OTUs between training and test datasets. The output of this step includes two simulated training datasets and one test dataset. B: Naive and ComBat normalization settings. In the Naive setting, the output datasets from Step A are directly used without any additional normalization. In the ComBat normalization setting, the two training datasets from Step A undergo normalization using the ComBat method to address potential batch effects by using the test dataset as a reference. C: Classification stage of applying different ensemble weighted learning and rank aggregation methods. The two training datasets from Step B are used to train two machine learning classifiers. These classifiers generate two lists of prediction probabilities when applied to the test dataset. For ensemble weighted learning, the two lists of probabilities are directly integrated using specific weights $(w_1, w_2)$ determined by the integration method, resulting in a final list of prediction probabilities. For rank aggregation, the two lists of probabilities are ranked, and the resulting lists of ranks are integrated using the respective rank aggregation methods. Different weights $(a_1, a_2)$ determined by the method are used in this integration process, producing one final list of ranks. Note that the integration methods utilize distinct weights $(w_1, w_2, a_1, a_2)$ based on their specific approach.

disease-associated OTUs between training and test datasets are varied. This constitutes the third scenario in our research workflow. We adjusted the number of disease-related microbes that overlap in the training and test disease models, and then simulated different datasets in line with these adjustments. The methodology for this process is outlined in Scenario 3: Varying degrees of overlap in disease-associated OTUs between training and test datasets.

Following the creation of the training and test datasets from the three scenarios mentioned above, depicted in Fig 1A, our goal was to assess the efficacy of the merging and integration methods in two distinct settings, as demonstrated in Fig 1B. These settings are: the naive setting, which involves the direct utilization of the simulated training and test datasets from the initial step into the third classification phase; and the ComBat normalization setting, which involves normalizing the two training datasets prior to the third classification phase. The process by which we conducted ComBat normalization on the training datasets is discussed in detail in Naive and ComBat normalization stage.

Subsequent to the previous steps, the two simulated training datasets and one test dataset are used as input for the final classification step of in this workflow. At this stage, we independently train a machine learning classifier on each training dataset and subsequently apply each to the test dataset. This generates two distinct lists of prediction probabilities. Then, we apply various integration methods to these two probability lists to yield the final results.

The two lists of prediction probabilities from the trained machine learning classifiers are directly employed in ensemble weighted learning methods. However, when we apply rank aggregation methods, the prediction probabilities are converted into ranks. A comprehensive explanation of the ensemble weighted learning and rank aggregation methods we used can be found in Classification stage.

In addition to these integration methods, we also adopted a merging strategy. As illustrated in Fig 1C, we combined the two training datasets and trained a machine learning classifier on this merged dataset. This trained classifier then made predictions directly on the test dataset to yield final results. The performances of these merging and integration methods are compared in the following sections.

In the subsequent three sections, we delve into the specifics of the three stages within our workflow: the Simulation Stage, the Naive and ComBat Normalization Stage, and the Classification Stage. Each of these stages plays a crucial role in our analyses, and understanding them is fundamental to comprehending the entire process.

## Simulation stage

**Scenario 1: Divergent training and test populations with varied genomic feature distribution.**   In the first scenario, we contemplated a situation where the two training populations are distinct from each other and both are different from the test population. We adjusted our simulations to reflect the extent of these differences. To emulate this scenario, we first established the baseline abundance levels for the operational taxonomic unit (OTU) across different populations.

Three probability vectors were created, each representing the underlying OTU abundance levels in three distinct populations. These were adapted from three real colorectal cancer (CRC) metagenomic datasets. We curated a total of six publicly accessible, geographically diverse CRC metagenomic datasets with download links provided in their respective original papers [5, 20–24]. We excluded samples from adenoma patients, only using samples from CRC patients and healthy controls. The case-control numbers and countries of origin for each dataset are displayed in Table 1.

**Table 1. Overview of six colorectal cancer-related metagenomic datasets.**

| Dataset | Country | No. of cases | No. of controls | Reference |
|---------|---------|--------------|-----------------|-----------|
| Zeller | France | 91 | 93 | [20] |
| Yu | China | 74 | 54 | [21] |
| Hannigan | USA/Canada | 27 | 28 | [22] |
| Feng | Austria | 46 | 63 | [23] |
| Vogtmann | USA | 52 | 52 | [24] |
| Thomas | Italy | 61 | 52 | [5] |

We created a PCoA plot to illustrate the population differences among these six CRC datasets, as shown in Fig 2. Based on this plot, we selected the three least overlapping populations—those represented by the Hannigan, Feng, and Yu datasets—as the foundation for generating the background OTU relative abundance vectors. The Hannigan and Yu datasets were utilized to generate training data, while the Feng dataset was used for test data generation.

The two training datasets were pre-processed to preserve the top 1000 OTUs with the largest variance in each dataset. We then amalgamated the OTUs from the two datasets to create a comprehensive set of OTUs for the subsequent analysis. This simulation study incorporated a total of 1,267 OTUs. We maintained the OTU count of the 1,267 OTUs for the Feng dataset, which forms the background distribution for simulating the test dataset, and removed the remaining OTUs. Subsequently, the count data was converted into a relative abundance vector, calculated by dividing each OTU's total count from all samples by the total counts of all OTUs.

Let us denote $v_1$, $v_2$, and $v_3$ as the three background relative abundance vectors derived from the microbial abundance profiles of the healthy control samples in the pre-processed Hannigan, Yu, and Feng datasets, respectively. These three vectors share the same 1,267 dimensionality, with each dimension representing the relative abundance level for each OTU. In order to scrutinize the influence of population differences on cross-study prediction, we devised a pseudo-population with a relative abundance vector defined as follows:

$$v_1(\alpha) = \alpha v_1 + (1 - \alpha)v_2 \tag{1}$$

Note that $v_1(\alpha) - v_2 = \alpha(v_1 - v_2)$. Consequently, the difference between the two populations simulated based on $v_1(\alpha)$ and $v_2$ escalates with $\alpha$. When $\alpha = 0$, both simulated populations share the same underlying distribution, hence eliminating population differences between the two training sets. Conversely, at $\alpha = 1$, the two simulated populations exhibit the maximum difference. We employed diverse $\alpha$ values, ranging from 0 to 1 in increments of 0.2, to reflect varying degrees of training population differences in subsequent analyses. The relative abundance profiles $v_1(\alpha)$ and $v_2$ served as background relative abundance vectors for training populations, while $v_3$ was applied to the test population.

From these 1,267 OTUs, we randomly selected 10 OTUs, assuming these were associated with a specific disease of interest. Given that disease-associated OTUs can either be enriched or depleted, we presumed the first 5 OTUs to be enriched and the remaining 5 to be depleted. These 10 OTUs were consistent across all subsequent experiments. To quantify the disease effect on these associated OTUs, we defined a disease effect factor, $e_d$, and hypothesized that the relative abundance of these OTUs could be represented as:

$$\{relative\ abundance\}_{enriched} = \{relative\ abundance\} * e_d \tag{2}$$

$$\{relative\ abundance\}_{depleted} = \frac{\{relative\ abundance\}}{e_d} \tag{3}$$

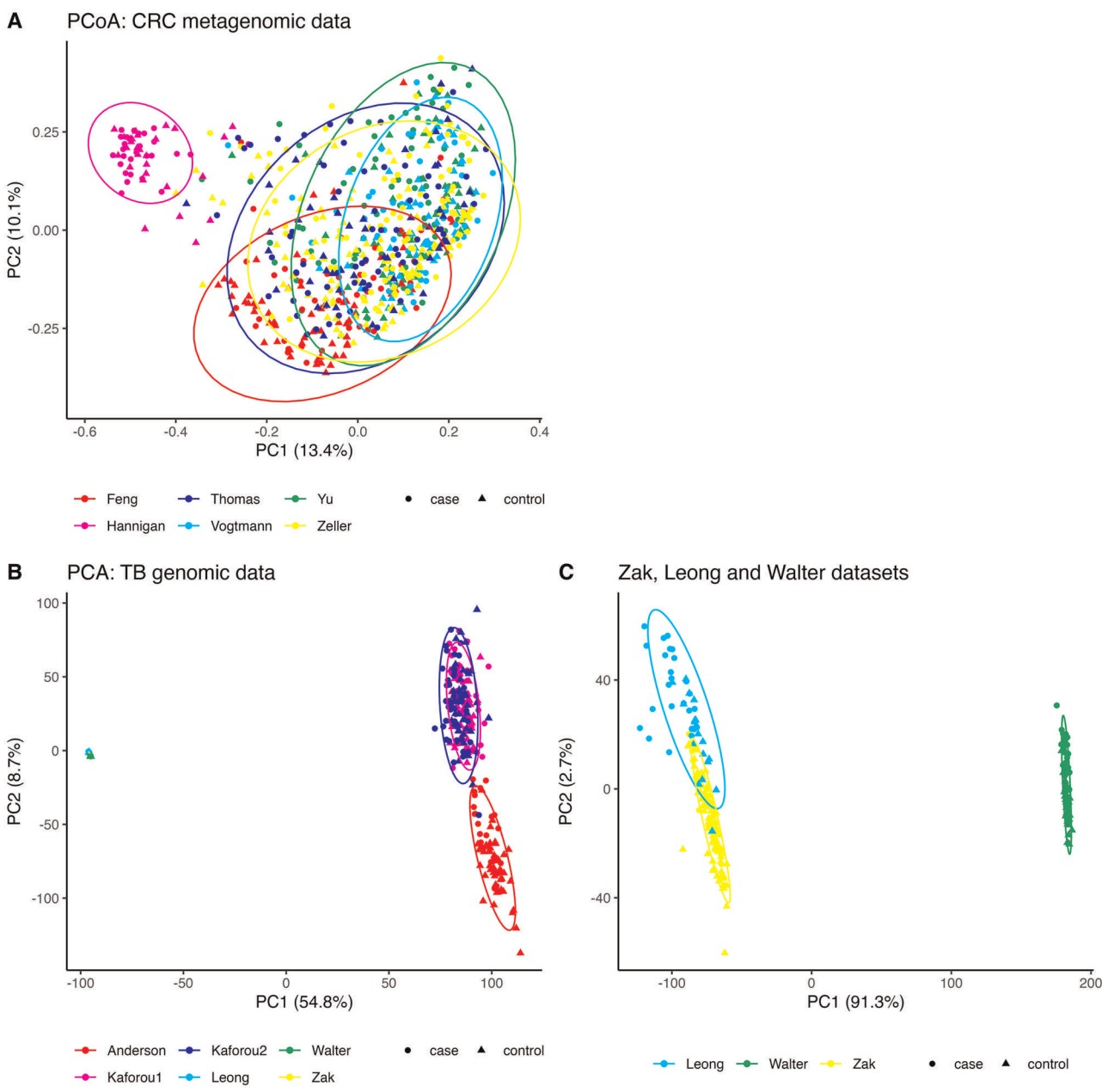

**Fig 2. Distributions of genomic features across multiple colorectal cancer and tuberculosis studies.** A: Principal Coordinate Analysis (PCoA) of Bray–Curtis distances, calculated from six colorectal cancer metagenomic count datasets. B: Principal Component Analysis (PCA) performed on six tuberculosis gene expression datasets. The PCA was carried out on the logarithm of fragments per kilobase of transcript per million mapped reads (log FPKM). In this figure, the Zak, Leong, and Walter datasets overlap with each other and are far away from the other three datasets. C: PCA performed on the Zak, Leong, and Walter tuberculosis gene expression datasets. The ellipses represent a 95% confidence level under the assumption of a multivariate t-distribution. A round dot represents a case sample, while a triangle dot signifies a control sample.

First, we adjusted $v_1(\alpha)$, $v_2$, and $v_3$ in this manner and normalized them into probability vectors, denoted as $v_1(\alpha)'$, $v_2'$, and $v_3'$, respectively. These were used to simulate case microbiome profiles. We adjusted $e_d$ to be 1.05, 1.075, and 1.1 in our simulation studies to assess the impact of disease effect. The larger the $e_d$ value, the more pronounced the difference between case and control samples.

OTU pseudo counts for controls and cases in each population were simulated as follows. A library size of one million reads ($1 \times 10^6$) was used for all subsequent simulations. OTU pseudo counts were generated using a multinomial distribution $MN(1 \times 10^6, v)$ where $v$ denotes the relative abundance vector. For control sample simulations, $v_1(\alpha)$, $v_2$, and $v_3$ were utilized; for case simulations, $v_1(\alpha)'$, $v'_2$, and $v'_3$ were employed. Fifty controls and fifty cases were generated for each $v$, and the resultant datasets were denoted as *training*1, *training*2, and *test*, respectively. Subsequently, count data were transformed into log-transformed relative abundance data. This was achieved by first dividing the sample counts, followed by applying a zero-replacement strategy proposed by Martn-Fernndez et al. [25]. In this strategy, we identified the minimum non-zero abundance in the dataset, and replaced all zero abundances with 0.65 times this minimum non-zero abundance. Finally, the non-zero relative abundance data were log-transformed and used in all subsequent analyses.

**Scenario 2: Different batch effects on training data with consistent underlying population genomic feature distribution.** In this simulation, we kept the underlying populations identical for both the training and test data. The background OTU abundance profiles were selected from Yu et al. [21] as displayed in Table 1. This ensures no population variations as described in Scenario 1 between the training and test samples. We designated 10 OTUs as disease-associated and set the library size to one million reads, with a disease effect factor $e_d$ fixed at 1.025. For each of the two training datasets and the test dataset, we generated 50 case and 50 control samples.

To simulate batch effects on the training data, we followed procedures similar to those in Zhang et al. [8]. We treated the two training datasets as two distinct batches and simulated different batch effects within them. The model for generating batch effects was based on the linear model proposed in the ComBat batch correction method [12], which postulates an additive impact on the mean of normalized OTU abundances and a multiplicative impact on the variance. We chose three severity levels for the effect on the mean ($sev_{mean} \in 0, 3, 5$) and three levels for the effect on the variance ($sev_{var} \in 1, 2, 4$). Consequently, the model generated batch effects on the two batches, adjusting the mean to $mean - sev_{mean}$, $mean + sev_{mean}$, and the variance to $var/sev_{var}$, $var \times sev_{var}$. We only applied the batch effects to the training data, leaving the test dataset unaltered. The two training batches and the test dataset were named as *batch*1, *batch*2, and *test*, respectively. Here, *batch*1 and *batch*2 serve the same roles as *training*1 and *training*2 from the prior scenario.

**Scenario 3: Varying degrees of overlap in disease-associated OTUs between training and test datasets.** During the simulation of different disease models, unlike the previous two scenarios where we fixed 10 disease-associated OTUs in both the training and test datasets, we introduced a variable parameter termed 'overlapping OTUs' in the test dataset. While the 10 disease-associated OTUs remained constant in the training data, the number of such OTUs in the test data varied between 2, 4, 6, 8, and 10 from the pool of 10 disease-associated OTUs in the training data. As the number of overlapping disease-associated OTUs increases, the disease models in the training and test data converge. When the number of overlapping OTUs reaches 10 in the test data, the disease models in the training and test data are identical.

After determining the overlapping OTUs between the training and test disease models, we proceeded as per the previous two scenarios to simulate two training datasets, using the background OTU distribution from the CRC dataset by Yu et al. [21], and one test dataset, using the background OTU distribution from the dataset by Feng et al. [23]. The details about the two datasets are presented in Table 1. The three datasets were simulated with the following parameters: 100 samples comprising of 50 cases and 50 controls, one million reads, and a

disease effect factor $e_d$ of 1.075. The two training datasets and the test dataset were named *training*1, *training*2, and *test*, respectively. No batch effects were introduced in this simulation.

### Naive and ComBat normalization stage

As depicted in Fig 1B, two distinct settings were established post-simulation of datasets (Fig 1A): a naive setup and a ComBat normalization process. In the naive setup, machine learning classifiers were trained directly on the unprocessed training datasets. These classifiers were then used to make predictions on the test dataset, and the resulting predictions were synthesized using various integration techniques.

In contrast, the ComBat normalization setting involved using the test dataset as a reference to normalize the two training datasets independently via the ComBat method. This process resulted in two new datasets, "Training1_ComBat" and "Training2_ComBat", while leaving the test dataset unaltered. Additionally, a combined dataset, "Merged_ComBat", was created by pooling together "Training1_ComBat" and "Training2_ComBat", serving as a counterpart for comparison with the "Merged" dataset. Machine learning classifiers were also trained on these normalized datasets and make predictions on the test dataset respectively.

In Scenario 2, we adjusted the terminology slightly due to the introduction of two batches, each subject to different batch effects. Here, "Training1", "Training2", "Training1_ComBat" and "Training2_ComBat" were renamed as "Batch1", "Batch2", "Batch1_ComBat" and "Batch2_ComBat", respectively. Additionally, we developed three distinct merging methods:"NoBatchEffect", where machine learning classifiers were trained on the original merged dataset without any simulated batch effect; "Merged", where classifiers were trained on a dataset that was an amalgamation of "Batch1" and "Batch2" where the batch effects have been simulated on the two batches, and "Merged_ComBat", which involved training on a combined dataset of "Batch1_ComBat" and "Batch2_ComBat.

The machine learning classifiers were trained independently on these different datasets, and subsequently used to generate predictions on the test data. Following this step, these predictions were amalgamated by using ensemble weighted learning or rank aggregation methods, resulting in the final predictor.

### Classification stage

**Machine learning classifiers.** Our research incorporated four machine learning classifiers: random forests (RF), logistic regression with L1 regularization (LASSO), support vector machine (SVM), and extreme gradient boosting (XGBoost). In the results section, we primarily present findings from the RF classifiers in the main texts. The results from LASSO are shown in S3, S6, S9, S12, S15 and S16 Figs; the results from SVM are shown in S4, S7, S10, S13, S17 and S18 Figs; and the results from XGBoost are shown in S5, S8, S11, S14, S19 and S20 Figs. In conjunction with the two normalization settings, each of these classifiers was trained on the respective training datasets. All of the four classifiers were implemented using the 'caret' package in R [26].

The RF classifiers incorporated 1000 decision trees, and the 'mtry' parameter was fine-tuned using 10-fold cross-validation. We opted to use the 'ranger' method for the train function to reduce computational time compared to the 'rf' method. For the LASSO classifier, the regularization parameter was selected from a range of 0 to 1 in increments of 0.001. This parameter selection was made to maximize the area under the operational characteristic curve (AUC) as determined by 10-fold cross-validation. Regarding the SVM, we employed a polynomial kernel with a default cost parameter of 1, a default degree of 3, and a default gamma of

0.1. Lastly, for the XGBoost classifier, we set the maximum number of trees to be created at 1000 and tested the maximum depth of the tree over the values 2, 4, 6, 8, and 10.

After training these classifiers, they were applied to the test dataset to produce prediction probabilities. For certain integration methods that necessitate a validation dataset, we divided the test dataset randomly into two halves: 50% served as validation data (*val*), and the other 50% remained as test data (*test*). We ensured an even split between case and control samples to prevent any potential bias.

**Ensemble weighted learning methods.** Patil et al. [9] assessed the efficacy of cross-study learning by utilizing five alternative weighting approaches: a straightforward average of predictions from each single-study learner (referred to as "Avg"), an average weighted by the study's sample size ("n-Avg"), an average weighted by cross-study performance ("CS-Avg"), stacked regression ("Reg-s"), and the average of study-specific regression weights ("Reg-a"). Importantly, the latter three ensemble methodologies prioritized reproducibility.

In our study, we compared these five ensemble weighted learning methods. We also introduced two additional methods combined with an machine learning classifiers, as suggested in [13]. The specifics of these ensemble weighted learning methodologies are detailed below:

1. **Avg:** This approach simply takes the mean of the prediction probabilities for the test data, which were produced by classifiers trained on "Training1" and "Training2". The equation to calculate the prediction probability for sample $i$ in test data is shown below.

$$p_i = \frac{1}{2} \times (p_i^{training1} + p_i^{training2}) \tag{4}$$

2. **n_Avg:** This method calculates the weighted average based on the sample size of the test data prediction probabilities, with the weights originating from classifiers trained on "Training1" and "Training2". As the two training datasets in the simulations share the same sample sizes, "n_Avg" yields the same results as "Avg". However, in real data applications where the dataset sample sizes can vary, these two methods may deliver differing outcomes.

$$p_i = \frac{(n_1 \times p_i^{training1} + n_2 \times p_i^{training2})}{n_1 + n_2} \tag{5}$$

3. **CS-Avg:** This method involves taking an average weighted by cross-study performance. First, a classifier is trained on "Training1" and then used on the samples from "Training2" to generate prediction probabilities. From these probabilities, we calculate the cross-entropy loss, defined as follows:

$$cel = -\frac{1}{N} \sum_{i=1}^{N} y_i \log(p_i) + (1 - y_i)\log(1 - p_i), \tag{6}$$

where $N$ represents the sample size of the test dataset, $y_i$ is the actual case/control status of sample $i$ ($y_i$ equals 0 for a control sample, or 1 for a case sample), and $p_i$ denotes the prediction probability for sample $i$ being a case sample, as determined by the machine learning classifier.

Subsequently, we computed the cross-entropy loss of the classifier trained on "Training1" and predicting on "Training2", and denoted it as $cel_1$. Similarly, the classifier trained on "Training2" predicting on "Training1" yielded a cross-entropy loss termed $cel_2$. We then calculated the weights as $weight_1 = |cel_1 - max(cel_1, cel_2)|$, and $weight_2 = |cel_2 - max(cel_1, cel_2)|$, and normalized these two weights to a sum of 1 then name them as $weight_1^{norm}$ and $weight_2^{norm}$. The

prediction probability for sample $i$ in test data is then calculated as:

$$p_i = weight_1^{norm} \times p_i^{training1} + weight_2^{norm} \times p_i^{training2} \tag{7}$$

Notably, in the context of two training datasets, the "CS-Avg" method always assigns a zero weight to the classifier exhibiting the worst cross-study performance (i.e., higher cross-entropy loss). As a result, only one of the prediction probabilities from the classifiers trained on "Training1" and "Training2" is utilized in this method under simulation conditions. In a broader scenario where multiple training datasets are present, the cross-entropy loss for each training dataset predicting on all other datasets is calculated. For instance, $cel_{ij}$ is calculated using prediction probabilities for samples in dataset $j$, trained on dataset $i$. The total cross-entropy loss for dataset $i$ is then calculated as $cel_i = \sum_{j,j \neq i} cel_{ij}$. Similar to the two training dataset scenario, weights for each dataset are calculated as $weight_i = |cel_i - max(cel_1, cel_2, \ldots, cel_i, \ldots, cel_n)|$, where $n$ represents the total number of training datasets used. The final weights are normalized to sum to 1. This method assigns zero weight to the model that exhibits the worst average performance across all other training datasets.

4. **Reg-a:** This method involves the averages of study-specific regression weights, which are computed using non-negative least squares. The machine learning classifiers were trained separately on "Training1" and "Training2". Predictions were made on "Training1" and "Training2", yielding four lists of probabilities. When testing on "Training1", we fitted non-negative least squares to the two associated probability vectors, with the actual case/control status in "Training1" serving as the response, and we obtained two coefficients. The same procedure was repeated for testing on "Training2". This resulted in a $2 \times 2$ coefficient matrix, with each row representing test data and each column representing training data. Next, we multiplied the coefficients in each row by the sample size of the test data, and the weights were finally computed as the column average of the adjusted coefficients.

5. **Reg-s:** This method involves stacked regression weights, which are computed using non-negative least squares. The two prediction probability vectors obtained from the "Reg-a" method were stacked into a single vector for each test dataset. We then fitted non-negative least squares to the stacked vectors, with the actual case/control status serving as the response. The coefficients were then used as weights.

6. **val-auc:** Machine learning classifiers were trained independently on "Training1" and "Training2". We then applied these trained classifiers on validation data for prediction. The AUC scores were calculated by comparing the prediction probabilities to the actual disease status from samples in validation data, name them as $AUC_1^{val}$ and $AUC_2^{val}$. We then assign the two AUCs as weights to combine the test data prediction probabilities from the two trained classifiers. The equation to calculate the prediction probability for sample $i$ in test data is shown below.

$$p_i = \frac{AUC_1^{val} \times p_i^{training1} + AUC_2^{val} \times p_i^{training2}}{AUC_1^{val} + AUC_2^{val}} \tag{8}$$

7. **LOSO-auc:** This method was proposed in our previous study [13] which involves the Leave-One-Sample-Out (LOSO) AUCs calculations. The high level idea is to assign different weights to prediction probabilities from classifiers trained on "Training1" and "Training2" by their respective prediction accuracy without bringing any inference from the test data. For each sample $i$ in the test data, we obtained two prediction probabilities $p_i^{training1}$ and $p_i^{training2}$ by applying the two classifiers trained on the two training datasets to the sample $i$

respectively. We then excluded sample *i* from the test data and computed AUC scores $AUC_i^{training1}$ and $AUC_i^{training2}$ by comparing the prediction probabilities from the rest of samples in the test data to their ground truth disease statuses. Then, the LOSO method assign the corresponding $max(AUC_i^{training1} - 0.5, 0)$ and $max(AUC_i^{training2} - 0.5, 0)$ as weights to combine $p_i^{training1}$ and $p_i^{training2}$ to get a final prediction probabilty for sample *i*.

$$p_i = \frac{max(AUC_i^{training1} - 0.5, 0) \times p_i^{training1} + max(AUC_i^{training2} - 0.5, 0) \times p_i^{training2}}{max(AUC_i^{training1} - 0.5, 0) + max(AUC_i^{training2} - 0.5, 0)} \tag{9}$$

In the ComBat normalization scenarios, these seven integration techniques were similarly applied to prediction probabilities acquired from classifiers trained on the "Training1_ComBat", "Training2_ComBat" and "Merged_ComBat" datasets. The performance of each method was assessed by calculating the AUC scores, which were derived by comparing the final prediction probabilities with the actual case/control status.

**Rank aggregation methods.** Rank aggregation methods have not traditionally been applied for phenotype prediction. In this study, we explored the application of rank aggregation for integrating different predictors, and evaluated their performance. For each independent prediction method, we initially sorted the samples in the test data based on their prediction probabilities of being classified as cases, with the order being in descending probability. This led to the generation of a ranked list, with the lowest rank assigned to items with ties. This process was repeated to create two ranked lists based on the predictions generated by two training classifiers. Following this, we employed various rank aggregation methods to create a unified, aggregated rank list for the samples.

We explored the use of five rank aggregation methods, comparing their performance both within this group, and against the previously discussed ensemble learning methods.

**1. mean:** This approach takes the average of the two ranked lists.

$$r_i = \frac{1}{2} \times (r_i^{training1} + r_i^{training2}) \tag{10}$$

**2. geometric mean:** This method calculates the geometric mean of the two ranked lists.

$$r_i = \sqrt{r_i^{training1} \times r_i^{training2}} \tag{11}$$

**3. Stuart rank aggregation:** The first step in this method is to normalize the ranks into rank ratios. The order statistics proposed by Stuart et al. [27] are then used to create an aggregated rank list. For computation, we used the 'RobustRankAggreg' package in R, specifying 'stuart' as the method [28].

**4. Robust rank aggregation (RRA):** Proposed by Kolde et al. [28], this method is also based on order statistics but has improved computational efficiency and statistical stability. For each item in the rank list, the algorithm looks at its position and compares this with a baseline case in which all preference lists are randomly shuffled. Each item is then assigned a P-value, indicating how much better its position in the ranked lists is than would be expected by chance. These P-values are then used to re-rank the list. We used the 'RobustRankAggreg' package in R to compute the results [28].

**5. Bayesian analysis of rank data with covariates (BARC):** This method, developed by Li et al. [29], is a Bayesian-based rank aggregation approach that incorporates information from covariates. Although covariates are not of concern in our study, we used their rank

aggregation method without involving any covariates. The method is available from https://github.com/li-xinran/BayesRankAnalysis.

After obtaining the aggregated rank lists using the aforementioned methods, we calculated the AUC scores to assess the performance of these rank aggregation methods. Similar to the ensemble weighted learning methods, these rank aggregation methods were applied separately to both the naive and ComBat normalization settings. The final AUC score is the fraction of pairs where the case sample is ranked higher than the control sample. A perfect ranking, where all case samples are ranked above all control samples, would give an AUC of 1. A completely random ranking would on average give an AUC of 0.5, as case and control samples would be equally likely to be ranked higher.

## Applications on real CRC metagenomic datasets

We applied the developed methods for merging and integration to six real-world metagenomic datasets from CRC studies. We selectively used samples from patients who had been diagnosed with CRC, excluding those with adenoma. Thus, the analysis was carried out on samples from CRC patients and healthy controls. Each dataset varied in terms of the number of case and control samples, country of origin, and associated references, all of which are detailed in Table 1.

The microbial count profiles for the six CRC datasets were generated using MicroPro [30]. Following generation, the count data was log-transformed into relative abundance data, using the same procedures as those described in the data pre-processing section of the simulation studies (Scenario 1: Divergent training and test populations with varied genomic feature distribution). We further pre-processed each dataset by retaining the top 1000 Operational Taxonomic Units (OTUs) with the largest variance.

We implemented a Leave-One-Dataset-Out (LODO) approach for these realistic data applications. In this setting, each of the six datasets was successively treated as test data, with the remaining five datasets serving as training data. For the training data, we created a comprehensive set of all OTUs, filling in zero abundance where a specific OTU was missing from a dataset. As in the simulation studies, we applied ComBat normalization to the five training datasets. However, the training data in the naive setting was left unchanged.

The next steps involved training the machine learning classifiers independently on each of the five training datasets, and subsequently predicting on the test dataset, followed by applying the integration methods to the resulting five lists of prediction probabilities. As for the merging method, the five training datasets were amalgamated into a single dataset, on which a single machine learning classifier was trained.

## Applications on TB gene expression datasets

In the application to TB gene expression studies, we utilized six annotated and pre-processed datasets, originally collected by Zhang et al. [8]. These datasets represent different geographical regions and vary in the numbers of cases and controls. Detailed information regarding the origin, case-control breakdown, and associated references is provided in Table 2.

These TB datasets had already undergone transformation into the logFPKMs. As such, we directly selected the top 1000 gene features with the largest variance from each dataset. By taking the union of these genes, we compiled a comprehensive set of features for subsequent analyses.

Similar to the approach for CRC datasets, we implemented the Leave-One-Dataset-Out strategy for these TB gene expression datasets. This involved treating each dataset in turn as

**Table 2. Overview of six tuberculosis-related gene expression datasets.**

| Dataset | Country | No. of cases | No. of controls | Reference |
|---|---|---|---|---|
| Zak | South Africa/Gambia | 16 | 104 | [31] |
| Anderson | South Africa/Malawi/Kenya | 20 | 50 | [32] |
| Leong | India | 25 | 19 | [3] |
| Walter | USA | 35 | 35 | [33] |
| Kaforou1 | South Africa | 46 | 48 | [34] |
| Kaforou2 | Malawi | 51 | 35 | [34] |

test data, with the remaining datasets serving as training data. Detailed procedures for this strategy are outlined in the section discussing applications to real CRC metagenomic datasets (Applications on real CRC metagenomic datasets).

## Results

### ComBat normalization is essential for heterogeneous populations

In the first scenario, we assessed the influence of divergent background operational taxonomic unit (OTU) distributions in the training samples on the predictive performance. The results are depicted in Fig 3. When the training and test data possess disparate background OTU

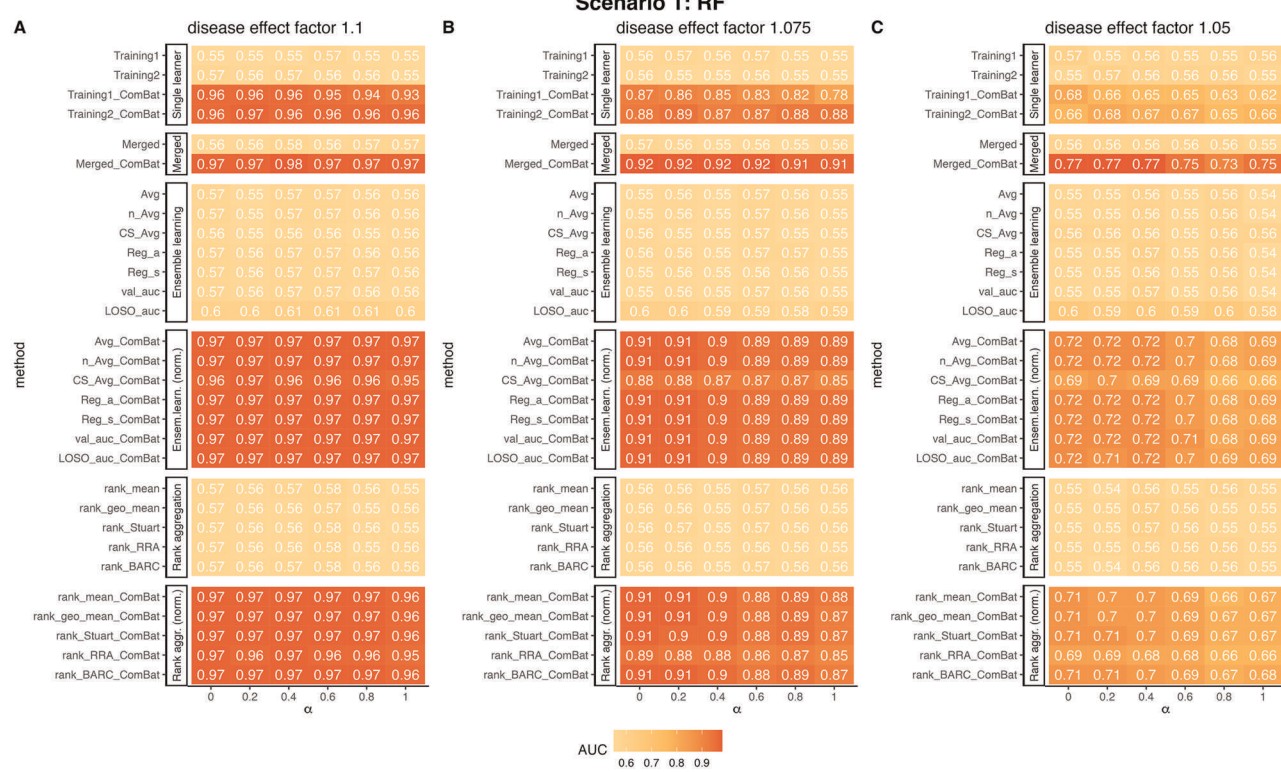

**Fig 3. ComBat normalization markedly enhances cross-study prediction when the training and test data have divergent feature distributions.** The figures display the AUCs of RF predictions employing various integration methods with three distinct disease effect factors. Columns represent different values of $\alpha$. Method names without a "ComBat" suffix refer to those implemented in the naive setting, while those with a "ComBat" suffix were conducted in the ComBat normalization setting. All the experiments were conducted 100 times, and the AUC scores presented in the figure represent the averages from these 100 trials. Abbreviations: Ensem.learn.-Ensemble Learning; Rank aggr.-Rank aggregation; norm.-normalization.

distributions, the direct application of the predictive model trained on the training data to the test data yields substantially low predictive accuracy with an AUC approximating 0.56, as illustrated in the "Training1" and "Training2" rows. Neither merging the raw training data nor directly integrating the trained models from multiple training samples enhanced the predictive accuracy. These findings underscore the critical role of normalization in the development of predictive models.

Many normalization methods have been developed for metagenomic data, but most of them primarily address experimental artifacts rather than population differences across studies ([12], [15], [16]). In our study, we applied ComBat [12], a widely used normalization method, to normalize the metagenomic data from different populations. We wanted to investigate if this normalization could improve prediction accuracy.

We observed that using ComBat to normalize the metagenomic data markedly enhanced the prediction accuracy in the test data. For example, the AUC score for "Training2_ComBat" increased from an average of 0.56 to approximately 0.96, 0.88, and 0.66 when the disease effect was set at 1.1, 1.075, and 1.05, respectively. We used the formula $v_1(\alpha) = \alpha v_1 + (1 - \alpha)v_2$ to calculate the background OTU relative abundance for "Training1", where $\alpha$ represents the difference between "Training1" and "Training2". We observed that the prediction accuracy decreased as $\alpha$ increased for all values of the disease effect $e_d$. For instance, when $e_d = 1.075$, the AUC for "Training1_ComBat" decreased from 0.87 to 0.78 as $\alpha$ increased from 0 to 1. This observation suggests that $v_1$ is further away from the test data compared to $v_2$, which could contribute to the decrease in prediction accuracy.

After observing the markedly increase in prediction accuracy for "Training1_ComBat" and "Training2_ComBat", we explored whether integrating the two predictors through ensemble weighted learning or rank aggregation could further enhance the prediction accuracy. The results, shown in the top rows marked in red, indicate that most integration methods performed similarly and outperformed both "Training1_ComBat" and "Training2_ComBat". Notably, the merging method after ComBat normalization, "Merged_ComBat", yielded the best performance. When the disease effect $e_d$ was relatively high (e.g., $e_d \geq 1.075$), the increase in AUC over other integration methods was minimal. However, when $e_d = 1.05$, the AUC for "Merged_ComBat" was 0.77, compared to approximately 0.72 for other integration methods when $\alpha$ was small. Similarly, the AUC for "Merged_ComBat" was 0.75, compared to about 0.69 for other integration methods when $\alpha$ was large.

These results indicate that normalizing for population differences before training machine learning models can play a crucial role in improving prediction performance. It is surprising considering that ComBat was originally designed to correct for experimental artifacts such as batch effects, not specifically for adjusting population differences. Our findings clearly demonstrate that ComBat can be used to adjust for population differences and enhance cross-study prediction accuracy.

## ComBat normalization effectively corrects batch effects within the same population

In the second scenario, we considered studies within the same population but conducted in different laboratories or using different sequencing technologies. In such cases, experimental batch effects can occur and it is crucial to correct these batch effects to ensure accurate and reliable predictions. As described in Scenario 2: Different batch effects on training data with consistent underlying population genomic feature distribution, we simulated batch effects that affected the mean and variance of OTU abundance levels, respectively. We evaluated the prediction accuracy of the RF classifier on the test data for each type of batch effect.

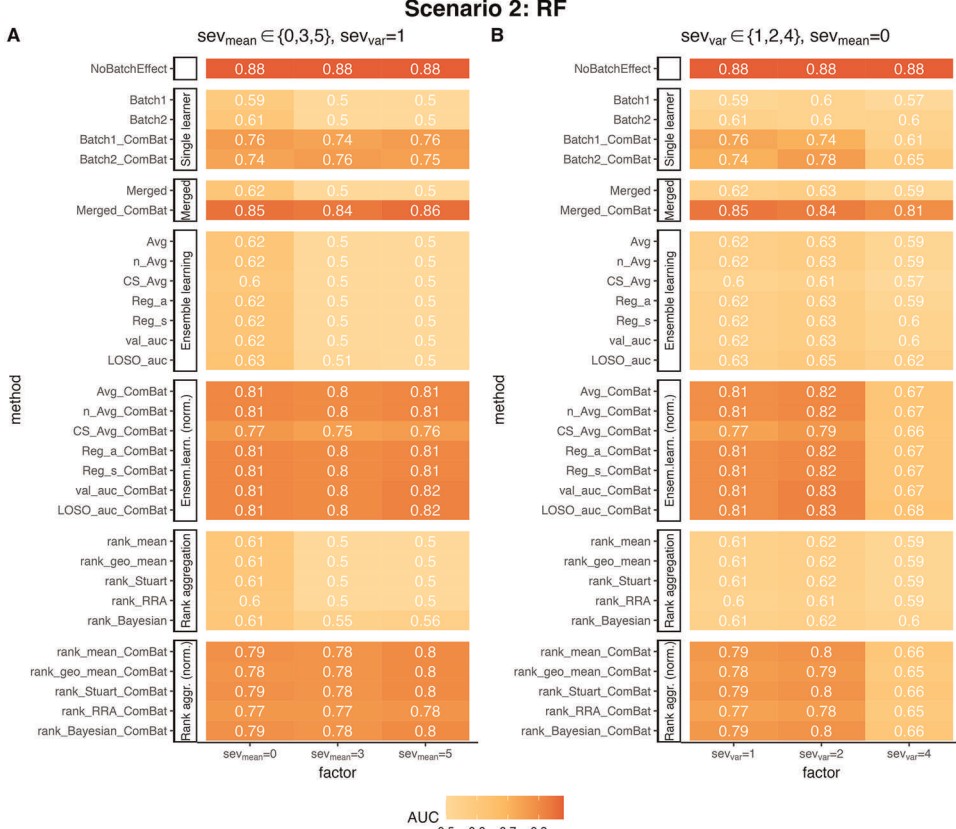

**Fig 4. ComBat normalization improves cross-study prediction in the presence of batch effects.** The figures show the AUC scores of RF prediction using different integration methods with varying severity levels of batch effects. A: AUC score comparisons with different severity levels of additive batch effects on the mean of OTU abundances, with no multiplicative batch effect on the variance. B: AUC score comparisons with different severity levels of multiplicative batch effects on the variance of OTU abundances, with no additive batch effect on the mean. The disease effect factor was set to 1.025 for both scenarios. The integration methods without a suffix of "ComBat" were applied in the naive setting, while those with a suffix of "ComBat" were applied after ComBat normalization. The experiments were repeated 100 times, and the shown AUC scores are the averages across the 100 trials. Abbreviations are the same as in Fig 3.

In this scenario, we set the disease effect $e_d = 1.025$ to clearly compare the performance of different methods. Fig 4A shows the results when there are additive effects on the mean ($sev_{mean} \in 0, 3, 5$) while the variance remains unchanged ($sev_{var} = 1$). Without data normalization, the AUC score on the test data is slightly higher than 0.5 when $sev_{mean} = 0$ and exactly 0.5 when $sev_{mean} \neq 0$, as expected. After applying ComBat normalization, the AUC scores on the test data increased to around 0.75 for all parameter values. Most of the ensemble weighted learning and rank aggregation methods applied further improved the AUC scores, approaching 0.8. However, it is worth noting that "rank_RRA_ComBat" and "CS_Avg_ComBat" slightly underperformed compared to other integration methods.

When simply merging the two training datasets after ComBat normalization, the highest AUC score of approximately 0.85 was achieved. This result demonstrates that the merging method, "Merged_ComBat", can effectively improve prediction accuracy even in the presence of batch effects within the same population. Furthermore, it is interesting to note that the prediction accuracy only slightly decreased from 0.88 to 0.85 when batch effects were absent, indicating the robustness and effectiveness of the "Merged_ComBat" method.

Fig 4B presents the results when there are multiplicative effects on the variance ($sev_{var} \in 1$, 2, 4) without any effect on the mean of OTU abundance levels ($sev_{mean} = 0$). In this case, without normalization, there is still some predictive power for the test data, although the AUC score is generally low at around 0.6 when considering the two batches separately. Integrating the two prediction probabilities without ComBat normalization does not improve the prediction accuracy.

However, when $sev_{var} = 1$ or 2, applying ComBat normalization improves the AUC scores based on the two training datasets, increasing them to approximately 0.75. Integrating the prediction probabilities after ComBat normalization further enhances the AUC to around 0.80. On the other hand, when $sev_{var} = 4$, ComBat normalization only increases the AUC to 0.67 when using ensemble weighted learning and rank aggregation methods. However, when merging the predictors after ComBat normalization, denoted as "Merged_ComBat", the highest AUC score of 0.81 is achieved.

Overall, these findings highlight the effectiveness of ComBat normalization in correcting batch effects within the same population and demonstrate the potential of the "Merged_ComBat" method to achieve high prediction accuracy.

## Prediction accuracy can be markedly decreased as the number of overlapping disease associated OTUs decreases

In Scenario 3, we examined the impact of varying disease models in the training and test data on the predictive accuracy of the Random Forest (RF) classifier. The results, presented in Fig 5, demonstrated a general consistency in the relative performance of diverse prediction methods with the findings from Scenarios 1 and 2, given that the number of overlapping disease-associated Operational Taxonomic Units (OTUs) exceeded four. However, in this context, the merging method did not clearly outperform other ensemble-weighted learning and rank aggregation methods under normalization setting. Despite this, it retained a commendable performance compared to alternative methods.

As expected, we observed a noticeable dip in prediction accuracy with the reduction in the count of overlapping disease-associated OTUs between the training and test data. For instance, when $e_d = 1.075$ and the overlap in disease-associated OTUs was fewer than 6, the AUC value was below 0.57. This value experienced an upswing to roughly 0.7 as the count of overlapping disease-associated OTUs escalated to between 6 and 8. Further, with an overlap of 10 disease-associated OTUs, the optimal AUC value reached approximately 0.91. These findings vividly illustrate the impact of disparities in disease models between training and test data on predictive performance. Additionally, when faced with significant variations between the training and test disease models, neither merging nor integration techniques were found beneficial, even when utilized in conjunction with normalization methods.

## Applications to metagenomic datasets related to colorectal cancer

We applied our methods to analyze six metagenomic datasets related to colorectal cancer (CRC) as summarized in Table 1. In this analysis, we employed a Leave-One-Dataset-Out (LODO) setting, where one dataset was treated as the test data and the other five datasets were used as training data. The methodology for this analysis is described in detail in Applications on real CRC metagenomic datasets. Fig 6A presents the average AUC scores from the six LODO experiments for the "Merged", "Ensemble weighting", "Ensemble weighting (normalized)", "Rank aggregation" and "Rank aggregation (normalized)" methods. The "Single learner" results in each LODO experiment represent the average performance of the five single

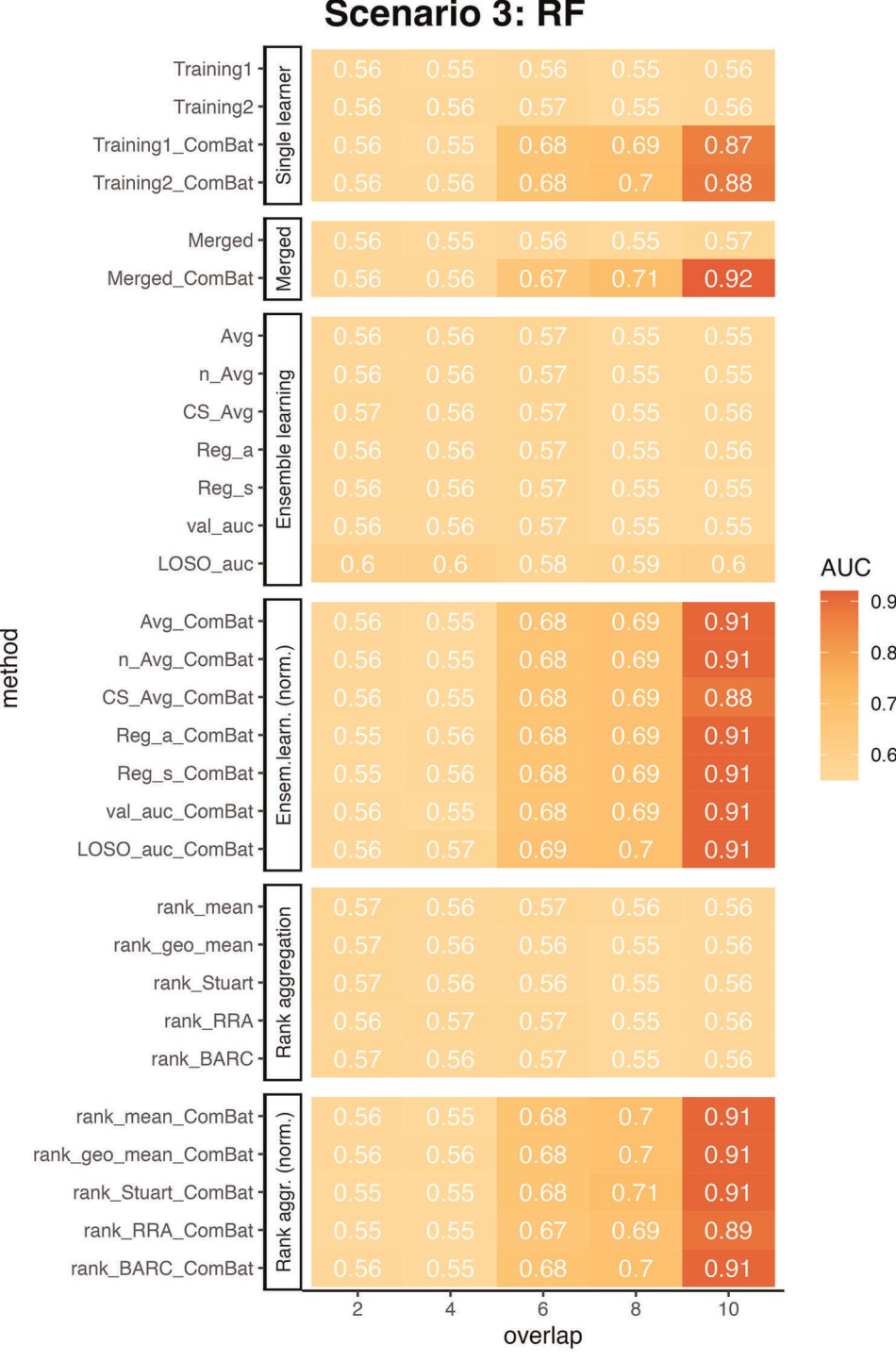

**Fig 5. ComBat normalization markedly increases cross-study prediction when the training and test data have varying degrees of overlap in disease-associated OTUs.** The figures show the AUCs of RF prediction using different integration methods with various number of overlapping disease associated OTUs. The disease effect factor was set to 1.075. Columns represent different numbers of overlapping disease associated OTUs in the training and test data, the larger the number, the more similar the two disease models are. When the number achieves 10, the two models are the same in the training and test data. All the experiments were repeated for 100 times and the AUC scores shown on the figure are the averages from the 100 trials. Abbreviations are the same as in Fig 3.

learners (trained on the five training datasets), followed by the average across the six LODO experiments. The individual results for each LODO experiment can be found in S1 Fig.

As shown in Fig 6A, using a single training dataset and predicting on the test data resulted in an average AUC increase from 0.63 to 0.65 when ComBat normalization was applied to the training data. The AUC scores for ensemble weighted learning and rank aggregation methods both showed slight improvements after ComBat normalization, but none of the integration methods outperformed merging the five training datasets. Comparing the prediction performance of the merging and integration methods with that of the single learner, it is evident that the AUC scores increased by approximately 0.1 on average, further supporting the notion that cross-study prediction using multiple training datasets is more accurate than relying on a single training dataset.

Examining the individual training results as shown in S1 Fig, we observed similar trends to those in Fig 6A in most cases. Among the six test datasets, the Hannigan dataset consistently exhibited the lowest AUC scores, and neither merging with ComBat normalization nor ensemble weighted learning methods improved the prediction performance. Additionally, the AUC scores when using the Hannigan dataset as the test data were the lowest among the six results, with an average of 0.61 for the single learner and 0.63 for the integration methods. This observation aligns with the data distribution depicted in the PCoA plot (Fig 2A), where the Hannigan dataset appears to be the most distinct and least overlapping with the other five datasets. This highlights the substantial differences in the count data distribution of the Hannigan dataset compared to the other five datasets. As demonstrated in our simulation studies, differences in the background distributions of genomic features among populations can impact the reproducibility of machine learning classifier's prediction performance.

## Applications to gene expression datasets related to tuberculosis

To further examine the prediction performance of merging and integration methods on real datasets, we utilized six TB gene expression datasets as summarized in Table 2 and followed the procedures described in Applications to metagenomic datasets related to colorectal cancer for the gene expression data.

In the TB studies application (Fig 6B), the overall AUC results were substantially higher than those observed in the CRC studies, and ComBat normalization led to significant improvements across all analyzed methods on average. Unlike the CRC studies, the ensemble weighted learning methods slightly outperformed the merging method in both the naive and ComBat normalization settings.

From the individual plots S2 Fig, we observed that when the Zak and Anderson datasets were used as test data, the prediction results were lower compared to the other four datasets, and the improvements resulting from ComBat normalization were smaller as well. This observation is consistent with the study by Zhang et al. [8], where these two datasets exhibited the highest cross-entropy losses when used as test data. One possible explanation for these differences is that the Zak and Anderson datasets only included children or adolescents, while the other four datasets also included adults. According to Alcaïs et al. [35], children and adults exhibit different tuberculosis clinical features and pathogenesis, which can influence the reproducibility of machine learning models when populations have distinct disease characteristics.

We also observed remarkable variations among training different single learners. For instance, in S2A Fig, when the RF classifier was trained on the Walter and Leong datasets and used to predict on the Zak dataset, the AUC results were much higher compared to training on Anderson, Kaforou1, and Kaforou2. Similarly, when training the RF classifier on Kaforou1 and Kaforou2 and predicting on Anderson, the AUCs reached 0.83 and 0.88, respectively,

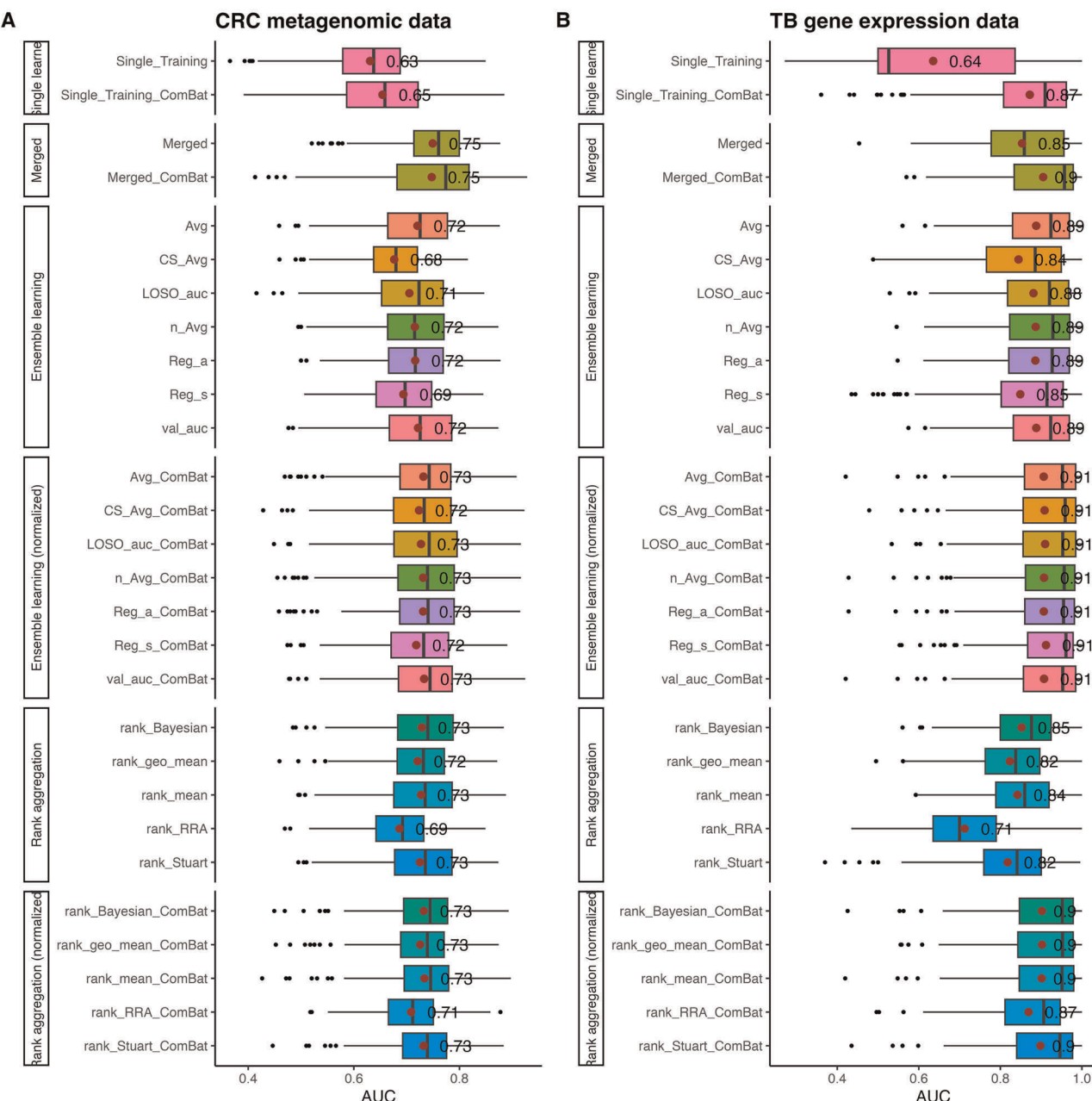

**Fig 6. Realistic applications of merging and integration methods on multiple CRC metagenomic datasets and TB gene expression datasets using RF classifiers with average AUC socres from six individual Leave-one-dataset-out(LODO) experiments.** A: Leave-one-dataset-out average AUC score comparisons among different methods in colorectal cancer metagenomic datasets. B: Leave-one-dataset-out average AUC score comparisons among different methods in tuberculosis gene expression datasets. The results by different methods are grouped into six groups. "Single learner": Each of the five training datasets were trained independently with RF classifier and predicted on the test dataset, then the average AUC score was taken among the five predictions. "Merged": Merging method with pooling all five training datasets into one training data. The "Single learner" and "Merged" experiments were conducted under both naive and ComBat normalization settings. "Ensemble learning": The five training predictors were integrated by ensemble weighted learning methods under naive setting. "Ensemble learning (normalized)": The five training predictors were integrated by ensemble weighted learning methods under ComBat normalization setting. "Rank aggregation": The five training predictors were integrated by rank aggregation methods under naive setting. "Rank aggregation (normalized)": The five training predictors were integrated by rank aggregation methods under ComBat normalization setting. The red dots and associated values on the figure are the mean AUC scores for each method, the vertical bars are the median AUC scores for each method, while the black dots represent the outliers. Same method under different settings are represented in the same color of boxplots. All the experiments were repeated 30 times for each test dataset, and the results presented in the figure were based on the average AUC scores of the total 180 replications for the six test datasets for metagenomic and gene expression studies, respectively.

while training on the other three datasets and predicting on Anderson resulted in an AUC of 0.5, resembling a random guess. These observations align with the data distribution in the PCA plots (Fig 2B), where Anderson, Kaforou1, and Kaforou2 are closer to each other and further from Zak, Walter, and Leong, while the latter three datasets are closer to each other. This further underscores the significant impact of dataset heterogeneity on the reproducibility of machine learning classifiers.

Finally, based on the results from the real applications in CRC and TB studies, consistent with the simulation study results mentioned earlier, we demonstrate that the ComBat normalization method is crucial for handling heterogeneous populations. When dealing with heterogeneous populations, it is recommended to employ both merging and integration methods to obtain the best prediction results.

## Consistency across LASSO, SVM, and XGBoost classifiers mirroring RF Classifier in simulation studies and real data applications

In addition to the RF classifier, we employed the LASSO, SVM, and XGBoost classifiers in experiments involving both simulations and real data applications. The trends observed for the case/control status prediction AUC values in both simulations and real data applications were similar across all classifiers.

In the simulations for Scenario 1, we noticed enhancements in AUC values using the ComBat normalization combined with merging and integration methods across all three classifiers. The improvements, however, were not as notable for the LASSO (S3 Fig) and SVM classifiers (S4 Fig) as they were for the RF (Fig 3) and XGBoost classifiers (S5 Fig).

During Scenario 2, we recorded noticeable improvements in AUC scores for the LASSO (S6 Fig) and XGBoost (S8 Fig) classifiers when we applied ComBat normalization along with merging and integration methods, particularly in the presence of batch effects that impacted the mean of OTUs abundances. The SVM classifier (S7 Fig) also demonstrated slight improvements under these conditions. Conversely, when batch effects perturbed the variance of OTU abundances, the XGBoost classifier showed a clear improvement in AUCs, while neither LASSO nor SVM classifiers demonstrated any evident enhancements.

In Scenario 3, the AUCs of the SVM (S10 Fig) and XGBoost (S11 Fig) classifiers improved when applying merging and integration methods along with normalization, given that the number of overlapping disease-associated OTUs exceeded four. However, we did not observe this advantage with the LASSO classifier (S9 Fig). Similarly, when there was a pronounced discrepancy in the disease models of the training and test data, none of the employed methods showed improved results across any of the classifiers.

In the CRC and TB applications, we observed a consistent performance across all methods and settings among the three additional classifiers (S12 to S20 Figs), mirroring the behavior of the RF classifier. While the enhancements in AUCs brought about by using ComBat normalization in combination with merging and integration methods were less pronounced compared to the simulations, they still indicated the potential for these methods to enhance the prediction performance of machine learning classifiers.

In comparing the four different classifiers, we observed that the RF and XGBoost classifiers delivered remarkably similar AUC results in both simulations and real data applications, with a much larger enhancement in the performance compared to LASSO and SVM when using normalization with merging and integration methods. In the simulation studies, the LASSO classifier delivered superior prediction performance compared to the other three classifiers. In contrast, the SVM classifier generally produced the least effective prediction performance. Overall, the outcomes from the three additional classifiers align closely with those from the RF

classifier, reinforcing the importance of applying normalization coupled with merging and integration methods. These strategies are key in mitigating the effects of heterogeneity and enhancing the reproducibility of machine learning classifiers.

## Rank aggregation is an alternative approach for integrating heterogeneous studies

Aggregating rank lists from multiple studies is a common strategy in genomic studies to gain a comprehensive understanding of biological phenomena. This approach provides a way to integrate heterogeneous data without the need for data normalization across studies. Previous studies have shown that aggregated rank lists yield more meaningful results than individual rank lists [27–29]. In our study, we extended this concept to integrate prediction probabilities from multiple machine learning classifiers by transforming them into rank lists and applying various rank aggregation methods. We focused on five established rank aggregation methods: mean of ranks, geometric mean of ranks, Stuart rank aggregation, Robust Rank Aggregation (RRA), and Bayesian Analysis of Rank data with Covariates (BARC). Detailed descriptions of these methods can be found in the Rank aggregation methods section.

In our simulation studies, we observed that the performance of the five rank aggregation methods was comparable to that of ensemble weighted learning methods in all three scenarios (Figs 3, 4 and 5). Notably, all five rank aggregation methods performed well when combined with ComBat normalization.

In the real applications of CRC and TB, the five rank aggregation methods showed slightly lower prediction performance compared to ensemble weighted learning methods in the naive settings. However, when combined with ComBat normalization, all five methods achieved similar and promising prediction results compared to ensemble weighted learning methods. Interestingly, S1 and S2 Figs demonstrate that for individual studies where ensemble weighted learning did not improve AUC scores (S1B and S1D Fig), the rank aggregation methods actually enhanced prediction performance compared to the ensemble weighted learning methods.

The simulations and real applications consistently highlight the value of aggregating rank lists of prediction probabilities when integrating multiple heterogeneous datasets for phenotype prediction. We have demonstrated that rank aggregation methods are as robust as ensemble weighted learning methods and can even improve prediction performance in cases where ensemble weighted learning does not yield good improvements.

## Discussion

With the increasing availability of large collections of omic data, the reproducibility of machine learning prediction models has raised great concerns when conducting cross-study predictions with the impact of study heterogeneity. Previous studies have addressed this issue and developed many statistical methods to overcome study heterogeneity, including merging with batch effect removal [36] and ensemble learning methods [9]. In this study, we performed a comprehensive analysis of different methods on the phenotype prediction by integrating heterogeneous omic studies. We considered three different sources of heterogeneity between datasets, including population differences, batch effects and different disease models. We developed a workflow in simulating these three sources of heterogeneity and generating simulated samples based on real datasets. We also evaluated the prediction performance of many different statistical methods, including merging, ensemble weighted learnings and rank aggregations. Besides the comparisons of different methods, we also explored the potential of normalizing the data by ComBat first then applied those statistical methods mentioned above. We

provided both simulation studies and real data applications on CRC metageomic datasets and TB gene expression datasets to compare different approaches.

In our simulation studies, we observed a decreasing trend in prediction accuracy among all statistical methods we investigated when the population heterogeneity became large, the batch effects increased on training data, as well as the differences of disease models between training and test data enlarged. These observations indicate that overcoming the heterogeneity needs to be addressed before applying machine learning prediction models on cross-study settings. Merging and integration methods that integrate different studies for phenotype predictions without batch correction did not improve the prediction accuracy much when compared to single training model, but when combined with ComBat normalization, we observed a remarkable improvement in the prediction accuracy in all simulations. These observations indicate normalizing the heterogeneous datasets before training machine learning models is essential in improving phenotype prediction performance.

It is noteworthy that our simulations yielded different conclusions in contrast with the study by Zhang et al. [8] on the prediction performance of using merging with ComBat normalization methods. In their study, they showed that merging with ComBat normalization was not as robust as ensemble learning methods at high severity of batch effects. However, in our second scenario of simulating different severity of batch effects on training and test data, we observed that merging combined with ComBat normalization always achieved highest prediction performance in spite of severity of batch effects (Scenario 2: Different batch effects on training data with consistent underlying population genomic feature distribution). We investigated the contradictions of observations in our study and the study by Zhang et al. [8], and we noticed that the ComBat normalization process in our study was different from theirs. In their study, when multiple training batches were provided, the batches were pooled into one training data, then applied ComBat normalization on the pooled dataset. They then did a second round of ComBat normalization on the test data using the pooled training data as reference. In our study, we normalized the training batches using the test data as reference when conducting ComBat normalization independently. The normalized training batches were pooled into one data for training machine learning model. Since test data is always the target of prediction, instead of adjusting the test data, we used it as the baseline to adjust different training batches so that the differences between training and test data were mitigated more effectively. Our study showed that our normalization approach yielded higher prediction accuracy.

Consistent with simulations, the applications on the CRC metagenomic and TB gene expression datasets with Leave-One-Dataset-Out experiments showed similar trends in terms of performance of merging and integration methods combined with ComBat normalization. However, the increasing trend in prediction accuracy is less remarkable than in the simulation studies. In our simulation studies, we intentionally manipulated the training and testing datasets as well as disease effect factors in the case and control samples to effectively highlight the improvements achieved by combining normalization and integration methods. In our simulation scenarios 1 and 3, the training and testing sets were chosen to be highly different. The disease effect in all the simulations was set to be relatively high. In contrast, in real-world scenarios, the disease effect factor may not be as pronounced as in our simulations, and the feature distribution between training and testing data maybe more similar than our simulation settings. In such situations, the differences among the results with/without batch normalization may not be as big as in the simulations. This observation is further corroborated by the comparisons under Scenario 1, where as the disease effect factor approaches 1 and thus decreases, the improvements in AUC scores achieved through the combination of normalization and integration methods become less noticeable. In contrast, when five training datasets were applied in real data applications as opposed to two in the simulations, all merging and

integration methods exhibited improved prediction performance, even in the absence of ComBat normalization. These findings underscore the value of integrating multiple studies, rather than relying solely on a single study, to bolster the reproducibility of machine learning models.

With the comparisons of the statistical methods used in our study, we saw similar trends for all the ensemble weighted learning methods except the slightly lower performance of the "CS-Avg" method. The "CS-Avg" method penalized the training dataset with the worst average performances on the rest of the training datasets when doing cross-training-data validations, and it excludes this dataset from predicting on test dataset. The worse performance of "CS-Avg" demonstrated that excluding the worst performance training data may not be beneficial to phenotype predictions as it discards useful information from that particular training data in the same time. Therefore, we suggest to use other ensemble weighted learning methods that also penalize worst performance training data but retain the useful information in some way. We also incorporated the rank aggregation methods into our study as well, and illustrated that the rank aggregation methods showed similar prediction performances, which also boosted the prediction accuracy remarkably. Rank aggregation methods should be considered as an alternative way for integrating heterogeneous studies in the future. We also noticed the extraordinary performance in merging method, and consistent with the findings by Guan et al. [37], merging and integration methods can outperform each other in different scenarios, and when training multiple studies we should consider to use both methods to find optimal.

Our findings derived from employing various machine classifiers parallel the trends observed with the use of RF classifiers in the main text for both simulation studies and real-world data applications. Intriguingly, we observed a large enhancement in the performance of tree-based classifiers, namely RF and XGBoost, when ComBat normalization coupled with integration methods were applied, while LASSO and SVM displayed only marginal improvements with these methodologies.

A limitation of the strategies in our study is that they require test data information for adjusting the heterogeneity effects from the training datasets. Consequently, when using a different test dataset, the training data needs to be readjusted and the machine learning classifiers need to be retrained. In comparison, methods like Cross-Platform Omics Prediction (CPOP) [38] are designed to make predictions on a single sample without normalization or the need for additional data integration. However, it should be noted that the CPOP method is developed with penalized regression classifier, and currently may not be compatible with machine learning methods, whereas our approach can be adapted with many different machine learning methods. In addition, CPOP can only be applicable to situations with a limited number of targeted features and can not be applicable to whole genomes or microbiomes.

## Conclusions

In conclusion, our study underscores the criticality of overcoming heterogeneity when incorporating different studies for phenotype prediction, an essential step in developing reproducible machine learning algorithms. Through the development of a comprehensive workflow, our research effectively simulates and evaluates various types of heterogeneity, integrating normalization method using ComBat and demonstrating its necessity for improving phenotype prediction performance. Our simulations and applications, applied to six colorectal cancer metagenomic studies and six tuberculosis gene expression studies, have underscored the substantial enhancement in prediction performance when normalization is coupled with a merging strategy and ensemble weighted learning methods. An additional key finding from our research is the potential of rank aggregation methods as an alternative approach to bolster prediction performance. Notably, these methods exhibit similar robustness as ensemble weighted

learning methods. These collective insights constitute an advancement in the field of phenotype prediction, offering concrete strategies to navigate and leverage the inherent heterogeneity in omics data, ultimately leading to more reliable and accurate outcomes.

## Supporting information

**S1 Fig. Realistic applications of merging and integration methods on multiple CRC metagenomic datasets using RF classifiers from six individual Leave-one-dataset-out(LODO) experiments.** The results by different methods are grouped into six groups. "Single learner": Each of the five training datasets were trained independently with RF classifier and predicted on the test dataset, then the average AUC score was taken among the five predictions. "Merged": Merging method with pooling all five training datasets into one training data. The "Single learner" and "Merged" experiments were conducted under both naive and ComBat normalization settings. "Ensemble learning": The five training predictors were integrated by ensemble weighted learning methods under naive setting. "Ensemble learning (normalized)": The five training predictors were integrated by ensemble weighted learning methods under ComBat normalization setting. "Rank aggregation": The five training predictors were integrated by rank aggregation methods under naive setting. "Rank aggregation (normalized)": The five training predictors were integrated by rank aggregation methods under ComBat normalization setting. The red dots and associated values on the figure are the mean AUC scores for each method, the vertical bars are the median AUC scores for each method, while the black dots represent the outliers. Same method under different settings are represented in the same color of boxplots. All the experiments were repeated 30 times for each test dataset. (TIF)

**S2 Fig. Realistic applications of merging and integration methods on multiple TB gene expression datasets using RF classifiers from six individual Leave-one-dataset-out(LODO) experiments.** The results by different methods are grouped into six groups. "Single learner": Each of the five training datasets were trained independently with RF classifier and predicted on the test dataset, then the average AUC score was taken among the five predictions. "Merged": Merging method with pooling all five training datasets into one training data. The "Single learner" and "Merged" experiments were conducted under both naive and ComBat normalization settings. "Ensemble learning": The five training predictors were integrated by ensemble weighted learning methods under naive setting. "Ensemble learning (normalized)": The five training predictors were integrated by ensemble weighted learning methods under ComBat normalization setting. "Rank aggregation": The five training predictors were integrated by rank aggregation methods under naive setting. "Rank aggregation (normalized)": The five training predictors were integrated by rank aggregation methods under ComBat normalization setting. The red dots and associated values on the figure are the mean AUC scores for each method, the vertical bars are the median AUC scores for each method, while the black dots represent the outliers. Same method under different settings are represented in the same color of boxplots. All the experiments were repeated 30 times for each test. (TIF)

**S3 Fig. ComBat normalization increases cross-study prediction when the training and test data have different feature distribution with LASSO classifiers.** The figures show the AUCs predictions of LASSO using different integration methods with three different disease effect factors. Columns represents different values of $\alpha$. All the method names without a suffix of "ComBat" are the methods carried out in the naive setting, while the names with a suffix of "ComBat" were carried out in the ComBat normalization setting. All the experiments were

repeated for 100 times and the AUC scores shown on the figure are the averages from the 100 trials. However, the differences between the results using normalization versus no-normalization are not as dramatic as other machine learning classifiers indicating robustness of LASSO with respect to population differences.
(TIF)

**S4 Fig. ComBat normalization showed similar low cross-study prediction accuracy compared to no normalization when the training and test data have different feature distribution with SVM classifiers.** The figures show the AUCs predictions of SVM with polynomial kernel using different integration methods with three different disease effect factors. Columns represents different values of $\alpha$. All the method names without a suffix of "ComBat" are the methods carried out in the naive setting, while the names with a suffix of "ComBat" were carried out in the ComBat normalization setting. All the experiments were repeated for 100 times and the AUC scores shown on the figure are the averages from the 100 trials.
(TIF)

**S5 Fig. ComBat normalization markedly increases cross-study prediction when the training and test data have different feature distribution with XGBoost classifiers.** The figures show the AUCs predictions of XGBoost using different integration methods with three different disease effect factors. Columns represents different values of $\alpha$. All the method names without a suffix of "ComBat" are the methods carried out in the naive setting, while the names with a suffix of "Com-Bat" were carried out in the ComBat normalization setting. All the experiments were repeated for 100 times and the AUC scores shown on the figure are the averages from the 100 trials.
(TIF)

**S6 Fig. ComBat normalization markedly increases cross-study prediction when the studies have batch effects when using LASSO classifiers.** The figures show the AUCs predictions of LASSO using different integration methods with various batch severity levels. A: AUC score comparisons with different severity levels of additive batch effects on the mean of OTU abundances, with no multiplicative batch effect on the variance. B: AUC score comparisons with different severity levels of multiplicative batch effects on the variance of OTU abundances, with no additive batch effect on the mean. The disease effect factor was set to 1.025 for both situations. All the method names without a suffix of "ComBat" are the methods done in naive setting, while the names with a suffix of "ComBat" were done in ComBat normalization setting. All the experiments were repeated for 100 times and the AUC scores shown on the figure are the averages from the 100 trials.
(TIF)

**S7 Fig. ComBat normalization slightly increases cross-study prediction accuracy when the studies have batch effects when using SVM classifiers.** The figures show the AUCs predictions of SVM with polynomial kernel using different integration methods with various batch severity levels. A: AUC score comparisons with different severity levels of additive batch effects on the mean of OTU abundances, with no multiplicative batch effect on the variance. B: AUC score comparisons with different severity levels of multiplicative batch effects on the variance of OTU abundances, with no additive batch effect on the mean. The disease effect factor was set to 1.025 for both situations. All the method names without a suffix of "ComBat" are the methods done in naive setting, while the names with a suffix of "ComBat" were done in Com-Bat normalization setting. All the experiments were repeated for 100 times and the AUC scores shown on the figure are the averages from the 100 trials.
(TIF)

**S8 Fig. ComBat normalization markedly increases cross-study prediction when the studies have batch effects when using XGBoost classifiers.** The figures show the AUCs predictions of XGBoost using different integration methods with various batch severity levels. A: AUC score comparisons with different severity levels of additive batch effects on the mean of OTU abundances, with no multiplicative batch effect on the variance. B: AUC score comparisons with different severity levels of multiplicative batch effects on the variance of OTU abundances, with no additive batch effect on the mean. The disease effect factor was set to 1.025 for both situations. All the method names without a suffix of "ComBat" are the methods done in naive setting, while the names with a suffix of "ComBat" were done in ComBat normalization setting. All the experiments were repeated for 100 times and the AUC scores shown on the figure are the averages from the 100 trials.
(TIF)

**S9 Fig. ComBat normalization markedly increases cross-study prediction when the training and test data have different disease models when using LASSO classifiers.** The figures show the AUCs predictions of LASSO using different integration methods with various number of overlapping disease associated OTUs. The disease effect factor was set to 1.075. Columns represent different numbers of overlapping disease associated OTUs in the training and test data, the larger the number, the more similar the two disease models are. When the number achieves 10, the two models are the same in the training and test data. All the experiments were repeated for 100 times and the AUC scores shown on the figure are the averages from the 100 trials.
(TIF)

**S10 Fig. ComBat normalization markedly increases cross-study prediction when the training and test data have different disease models when using SVM classifiers.** The figures show the AUCs predictions of SVM with polynomial kernel using different integration methods with various number of overlapping disease associated OTUs. The disease effect factor was set to 1.075. Columns represent different numbers of overlapping disease associated OTUs in the training and test data, the larger the number, the more similar the two disease models are. When the number achieves 10, the two models are the same in the training and test data. All the experiments were repeated for 100 times and the AUC scores shown on the figure are the averages from the 100 trials.
(TIF)

**S11 Fig. ComBat normalization markedly increases cross-study prediction when the training and test data have different disease models when using XGBoost classifiers.** The figures show the AUCs predictions of XGBoost using different integration methods with various number of overlapping disease associated OTUs. The disease effect factor was set to 1.075. Columns represent different numbers of overlapping disease associated OTUs in the training and test data, the larger the number, the more similar the two disease models are. When the number achieves 10, the two models are the same in the training and test data. All the experiments were repeated for 100 times and the AUC scores shown on the figure are the averages from the 100 trials.
(TIF)

**S12 Fig. Realistic applications of merging and integration methods on multiple CRC metagenomic datasets and TB gene expression datasets using LASSO classifiers with average AUC socres from six individual Leave-one-dataset-out(LODO) experiments.** A: Leave-one-dataset-out average AUC score comparisons among different methods in colorectal cancer metagenomic datasets. B: Leave-one-dataset-out average AUC score comparisons among

different methods in tuberculosis gene expression datasets. The results by different methods are grouped into six groups. "Single learner": Each of the five training datasets were trained independently with RF classifier and predicted on the test dataset, then the average AUC score was taken among the five predictions. "Merged": Merging method with pooling all five training datasets into one training data. The "Single learner" and "Merged" experiments were conducted under both naive and ComBat normalization settings. "Ensemble learning": The five training predictors were integrated by ensemble weighted learning methods under naive setting. "Ensemble learning (normalized)": The five training predictors were integrated by ensemble weighted learning methods under ComBat normalization setting. "Rank aggregation": The five training predictors were integrated by rank aggregation methods under naive setting. "Rank aggregation (normalized)": The five training predictors were integrated by rank aggregation methods under ComBat normalization setting. The red dots and associated values on the figure are the mean AUC scores for each method, the vertical bars are the median AUC scores for each method, while the black dots represent the outliers. Same method under different settings are represented in the same color of boxplots. All the experiments were repeated 30 times for each test dataset, and the results presented in the figure were based on the average AUC scores of the total 180 replications for the six test datasets for metagenomic and gene expression studies, respectively.
(TIF)

**S13 Fig. Realistic applications of merging and integration methods on multiple CRC metagenomic datasets and TB gene expression datasets using SVM classifiers with average AUC socres from six individual Leave-one-dataset-out(LODO) experiments.** A: Leave-one-dataset-out average AUC score comparisons among different methods in colorectal cancer metagenomic datasets. B: Leave-one-dataset-out average AUC score comparisons among different methods in tuberculosis gene expression datasets. The results by different methods are grouped into six groups. "Single learner": Each of the five training datasets were trained independently with RF classifier and predicted on the test dataset, then the average AUC score was taken among the five predictions. "Merged": Merging method with pooling all five training datasets into one training data. The "Single learner" and "Merged" experiments were conducted under both naive and ComBat normalization settings. "Ensemble learning": The five training predictors were integrated by ensemble weighted learning methods under naive setting. "Ensemble learning (normalized)": The five training predictors were integrated by ensemble weighted learning methods under ComBat normalization setting. "Rank aggregation": The five training predictors were integrated by rank aggregation methods under naive setting. "Rank aggregation (normalized)": The five training predictors were integrated by rank aggregation methods under ComBat normalization setting. The red dots and associated values on the figure are the mean AUC scores for each method, the vertical bars are the median AUC scores for each method, while the black dots represent the outliers. Same method under different settings are represented in the same color of boxplots. All the experiments were repeated 30 times for each test dataset, and the results presented in the figure were based on the average AUC scores of the total 180 replications for the six test datasets for metagenomic and gene expression studies, respectively.
(TIF)

**S14 Fig. Realistic applications of merging and integration methods on multiple CRC metagenomic datasets and TB gene expression datasets using XGBoost classifiers with average AUC socres from six individual Leave-one-dataset-out(LODO) experiments.** A: Leave-one-dataset-out average AUC score comparisons among different methods in colorectal cancer metagenomic datasets. B: Leave-one-dataset-out average AUC score comparisons among

different methods in tuberculosis gene expression datasets. The results by different methods are grouped into six groups. "Single learner": Each of the five training datasets were trained independently with RF classifier and predicted on the test dataset, then the average AUC score was taken among the five predictions. "Merged": Merging method with pooling all five training datasets into one training data. The "Single learner" and "Merged" experiments were conducted under both naive and ComBat normalization settings. "Ensemble learning": The five training predictors were integrated by ensemble weighted learning methods under naive setting. "Ensemble learning (normalized)": The five training predictors were integrated by ensemble weighted learning methods under ComBat normalization setting. "Rank aggregation": The five training predictors were integrated by rank aggregation methods under naive setting. "Rank aggregation (normalized)": The five training predictors were integrated by rank aggregation methods under ComBat normalization setting. The red dots and associated values on the figure are the mean AUC scores for each method, the vertical bars are the median AUC scores for each method, while the black dots represent the outliers. Same method under different settings are represented in the same color of boxplots. All the experiments were repeated 30 times for each test dataset, and the results presented in the figure were based on the average AUC scores of the total 180 replications for the six test datasets for metagenomic and gene expression studies, respectively.

(TIF)

**S15 Fig. Realistic applications of merging and integration methods on multiple CRC metagenomic datasets using LASSO classifiers from six individual Leave-one-dataset-out (LODO) experiments.** The results by different methods are grouped into six groups. "Single learner": Each of the five training datasets were trained independently with RF classifier and predicted on the test dataset, then the average AUC score was taken among the five predictions. "Merged": Merging method with pooling all five training datasets into one training data. The "Single learner" and "Merged" experiments were conducted under both naive and ComBat normalization settings. "Ensemble learning": The five training predictors were integrated by ensemble weighted learning methods under naive setting. "Ensemble learning (normalized)": The five training predictors were integrated by ensemble weighted learning methods under ComBat normalization setting. "Rank aggregation": The five training predictors were integrated by rank aggregation methods under naive setting. "Rank aggregation (normalized)": The five training predictors were integrated by rank aggregation methods under ComBat normalization setting. The red dots and associated values on the figure are the mean AUC scores for each method, the vertical bars are the median AUC scores for each method, while the black dots represent the outliers. Same method under different settings are represented in the same color of boxplots. All the experiments were repeated 30 times for each test dataset.

(TIF)

**S16 Fig. Realistic applications of merging and integration methods on multiple TB gene expression datasets using LASSO classifiers from six individual Leave-one-dataset-out (LODO) experiments.** The results by different methods are grouped into six groups. "Single learner": Each of the five training datasets were trained independently with RF classifier and predicted on the test dataset, then the average AUC score was taken among the five predictions. "Merged": Merging method with pooling all five training datasets into one training data. The "Single learner" and "Merged" experiments were conducted under both naive and ComBat normalization settings. "Ensemble learning": The five training predictors were integrated by ensemble weighted learning methods under naive setting. "Ensemble learning (normalized)": The five training predictors were integrated by ensemble weighted learning methods under ComBat normalization setting. "Rank aggregation": The five training predictors were

integrated by rank aggregation methods under naive setting. "Rank aggregation (normalized)": The five training predictors were integrated by rank aggregation methods under ComBat normalization setting. The red dots and associated values on the figure are the mean AUC scores for each method, the vertical bars are the median AUC scores for each method, while the black dots represent the outliers. Same method under different settings are represented in the same color of boxplots. All the experiments were repeated 30 times for each test.
(TIF)

**S17 Fig. Realistic applications of merging and integration methods on multiple CRC meta-genomic datasets using SVM classifiers from six individual Leave-one-dataset-out(LODO) experiments.** The results by different methods are grouped into six groups. "Single learner": Each of the five training datasets were trained independently with RF classifier and predicted on the test dataset, then the average AUC score was taken among the five predictions. "Merged": Merging method with pooling all five training datasets into one training data. The "Single learner" and "Merged" experiments were conducted under both naive and ComBat normalization settings. "Ensemble learning": The five training predictors were integrated by ensemble weighted learning methods under naive setting. "Ensemble learning (normalized)": The five training predictors were integrated by ensemble weighted learning methods under ComBat normalization setting. "Rank aggregation": The five training predictors were integrated by rank aggregation methods under naive setting. "Rank aggregation (normalized)": The five training predictors were integrated by rank aggregation methods under ComBat normalization setting. The red dots and associated values on the figure are the mean AUC scores for each method, the vertical bars are the median AUC scores for each method, while the black dots represent the outliers. Same method under different settings are represented in the same color of boxplots. All the experiments were repeated 30 times for each test dataset.
(TIF)

**S18 Fig. Realistic applications of merging and integration methods on multiple TB gene expression datasets using SVM classifiers from six individual Leave-one-dataset-out (LODO) experiments.** The results by different methods are grouped into six groups. "Single learner": Each of the five training datasets were trained independently with RF classifier and predicted on the test dataset, then the average AUC score was taken among the five predictions. "Merged": Merging method with pooling all five training datasets into one training data. The "Single learner" and "Merged" experiments were conducted under both naive and ComBat normalization settings. "Ensemble learning": The five training predictors were integrated by ensemble weighted learning methods under naive setting. "Ensemble learning (normalized)": The five training predictors were integrated by ensemble weighted learning methods under ComBat normalization setting. "Rank aggregation": The five training predictors were integrated by rank aggregation methods under naive setting. "Rank aggregation (normalized)": The five training predictors were integrated by rank aggregation methods under ComBat normalization setting. The red dots and associated values on the figure are the mean AUC scores for each method, the vertical bars are the median AUC scores for each method, while the black dots represent the outliers. Same method under different settings are represented in the same color of boxplots. All the experiments were repeated 30 times for each test.
(TIF)

**S19 Fig. Realistic applications of merging and integration methods on multiple CRC meta-genomic datasets using XGBoost classifiers from six individual Leave-one-dataset-out (LODO) experiments.** The results by different methods are grouped into six groups. "Single

learner": Each of the five training datasets were trained independently with RF classifier and predicted on the test dataset, then the average AUC score was taken among the five predictions. "Merged": Merging method with pooling all five training datasets into one training data. The "Single learner" and "Merged" experiments were conducted under both naive and ComBat normalization settings. "Ensemble learning": The five training predictors were integrated by ensemble weighted learning methods under naive setting. "Ensemble learning (normalized)": The five training predictors were integrated by ensemble weighted learning methods under ComBat normalization setting. "Rank aggregation": The five training predictors were integrated by rank aggregation methods under naive setting. "Rank aggregation (normalized)": The five training predictors were integrated by rank aggregation methods under ComBat normalization setting. The red dots and associated values on the figure are the mean AUC scores for each method, the vertical bars are the median AUC scores for each method, while the black dots represent the outliers. Same method under different settings are represented in the same color of boxplots. All the experiments were repeated 30 times for each test. (TIF)

**S20 Fig. Realistic applications of merging and integration methods on multiple TB gene expression datasets using XGBoost classifiers from six individual Leave-one-dataset-out (LODO) experiments.** The results by different methods are grouped into six groups. "Single learner": Each of the five training datasets were trained independently with RF classifier and predicted on the test dataset, then the average AUC score was taken among the five predictions. "Merged": Merging method with pooling all five training datasets into one training data. The "Single learner" and "Merged" experiments were conducted under both naive and ComBat normalization settings. "Ensemble learning": The five training predictors were integrated by ensemble weighted learning methods under naive setting. "Ensemble learning (normalized)": The five training predictors were integrated by ensemble weighted learning methods under ComBat normalization setting. "Rank aggregation": The five training predictors were integrated by rank aggregation methods under naive setting. "Rank aggregation (normalized)": The five training predictors were integrated by rank aggregation methods under ComBat normalization setting. The red dots and associated values on the figure are the mean AUC scores for each method, the vertical bars are the median AUC scores for each method, while the black dots represent the outliers. Same method under different settings are represented in the same color of boxplots. All the experiments were repeated 30 times for each test. (TIF)

## Acknowledgments

We thank Beibei Wang for advice on choosing and implementing batch normalization method.

## Author Contributions

**Conceptualization:** Fengzhu Sun.

**Data curation:** Yilin Gao.

**Formal analysis:** Yilin Gao.

**Funding acquisition:** Fengzhu Sun.

**Investigation:** Yilin Gao, Fengzhu Sun.

**Methodology:** Yilin Gao, Fengzhu Sun.

**Software:** Yilin Gao.

**Supervision:** Fengzhu Sun.

**Validation:** Yilin Gao.

**Visualization:** Yilin Gao.

**Writing – original draft:** Yilin Gao.

**Writing – review & editing:** Fengzhu Sun.

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
