## [Decision Letter · Decision Letter 0]

16 Jun 2023

Dear Dr. Sun,

Thank you very much for submitting your manuscript "Batch Normalization Followed by Merging Is Powerful for Phenotype Prediction Integrating Multiple Heterogeneous Studies" for consideration at PLOS Computational Biology.

We apologize for the long delay in getting back to you. This manuscript was particularly difficult to recruit reviewers for.

As with all papers reviewed by the journal, your manuscript was reviewed by members of the editorial board and by several independent reviewers. In light of the reviews (below this email), we would like to invite the resubmission of a significantly-revised version that takes into account the reviewers' comments.

The reviewers agree that there is interest in the study, but also raise important points which should be addressed before publication.

We cannot make any decision about publication until we have seen the revised manuscript and your response to the reviewers' comments. Your revised manuscript is also likely to be sent to reviewers for further evaluation.

Sincerely,

Luis Pedro Coelho

Academic Editor

PLOS Computational Biology

Edwin Wang

Section Editor

PLOS Computational Biology

The reviewers agree that there is interest in the study, but also raise important points which should be addressed before publication.

Reviewer's Responses to Questions

**Comments to the Authors:**

Reviewer #1: In this paper, Gao and Sun investigated different strategies phenotype prediction from multiple heterogeneous studies, including batch normalization, merging, ensemble and ranking aggregation. Through several simulation of three scenarios and real datasets, the authors found that using COMBAT normalization and mering can improve the phenotype prediction accuracy. This is an interesting study and here are my comments to the paper:

1. The authors only test one classifier in their study. I wonder whether the same observations applied to some other classifier, e.g. SVM, lasso and XGBoost etc.

2. The gap of the prediction accuracy rates between different approaches are much more dramatic in the simulated data compared to the real data. I am wondering how realistic the simulated datasets are. The authors discuss the potential reason is that as the number of training datasets increase, the gap between different approach will decrease, it will be good if the authors can explore this aspect in the simulation.

3. While batch normalization can remove the batch effect and study specific variation, the current and improve the phenotype prediction. The current strategies are limited as they require the test data information and thus the training data needs to be re-adjusted and the model needs to be retrained every time when a new test dataset comes in. It will be good that the author can discuss the limitation and also this approach compared to strategies like domain adaption and methods like CPOP (Wang et al. 2022).

• Wang, K.Y.X., Pupo, G.M., Tembe, V. et al. Cross-Platform Omics Prediction procedure: a statistical machine learning framework for wider implementation of precision medicine.npj Digit. Med. 5, 85 (2022). https://doi.org/10.1038/s41746-022-00618-5

Minor comments:

1. Figure 1c. To me this section is a “Classification” step, rather than an “Integration” step, as it contains model training + integration of results etc.

2. It will be good that the authors can include a conclusion paragraph at the end of the manuscript to summarize the key take home messages of the paper.

3. It will be good to group the different methods in Figure 3-5 like Figure 6.

Reviewer #2: The paper investigates the best approaches to integrate different studies of the same type of omics data under a variety of different heterogeneities. The authors developed a comprehensive workflow to simulate a variety of different types of heterogeneity and evaluate the performances of different integration methods together with batch normalization by using ComBat. They also demonstrated the results through realistic applications on six colorectal cancer (CRC) metagenomic studies and six tuberculosis (TB) gene expression studies, respectively. They showed that heterogeneity in different genomic studies can markedly negatively impact the machine learning classifier’s reproducibility. Albeit the idea is interesting and the results look good, I am not entirely convinced by the evidence presented in the paper

1.The authors indicated that the comBat normalization improved the prediction performance of machine learning classifier when heterogeneous populations presented, and could successfully remove batch effects within the same population. When “the methods” + “comBat”, the prediction performances were improved. But both “the methods” and “comBat” are existing methods, what is the biggest innovation of this study?

2.I tried to understand the method details, but failed. Please provide more details of the methods, especially for the newly developed part. Otherwise, it is hard to judge whether the new approach is worthwhile.

**Have the authors made all data and (if applicable) computational code underlying the findings in their manuscript fully available?**

Reviewer #1: None

Reviewer #2: Yes

PLOS authors have the option to publish the peer review history of their article (what does this mean?). If published, this will include your full peer review and any attached files.

Reviewer #1: No

Reviewer #2: No
---

## [Decision Letter · Decision Letter 1]

30 Sep 2023

Dear Dr. Sun,

We are pleased to inform you that your manuscript 'Batch Normalization Followed by Merging is Powerful for Phenotype Prediction Integrating Multiple Heterogeneous Studies' has been provisionally accepted for publication in PLOS Computational Biology.

Best regards,

Sushmita Roy, Ph.D.

Section Editor

PLOS Computational Biology

Sushmita Roy

Section Editor

PLOS Computational Biology

Reviewer's Responses to Questions

**Comments to the Authors:**

Reviewer #1: The authors have addressed my comments.

Reviewer #2: Thank you for revising the manuscript. I have no additional comments.

**Have the authors made all data and (if applicable) computational code underlying the findings in their manuscript fully available?**

Reviewer #1: None

Reviewer #2: None

PLOS authors have the option to publish the peer review history of their article (what does this mean?). If published, this will include your full peer review and any attached files.

Reviewer #1: No

Reviewer #2: No

---

## [Editor Report · Acceptance letter]

11 Oct 2023

PCOMPBIOL-D-22-01425R1 

Batch Normalization Followed by Merging is Powerful for Phenotype Prediction Integrating Multiple Heterogeneous Studies

Dear Dr Sun,

I am pleased to inform you that your manuscript has been formally accepted for publication in PLOS Computational Biology. Your manuscript is now with our production department and you will be notified of the publication date in due course.

With kind regards,

Zsofi Zombor
